# UNIFORM GENERALIZATION BOUNDS FOR OVERPARAMETERIZED NEURAL NETWORKS

## ABSTRACT

An interesting observation in artificial neural networks is their favorable generalization error despite typically being extremely overparameterized. It is well known that the classical statistical learning methods often result in vacuous generalization errors in the case of overparameterized neural networks. Adopting the recently developed Neural Tangent (NT) kernel theory, we prove uniform generalization bounds for overparameterized neural networks in kernel regimes, when the true data generating model belongs to the reproducing kernel Hilbert space (RKHS) corresponding to the NT kernel. Importantly, our bounds capture the exact error rates depending on the differentiability of the activation functions. In order to establish these bounds, we propose the information gain of the NT kernel as a measure of complexity of the learning problem. Our analysis uses a Mercer decomposition of the NT kernel in the basis of spherical harmonics and the decay rate of the corresponding eigenvalues. As a byproduct of our results, we show the equivalence between the RKHS corresponding to the NT kernel and its counterpart corresponding to the Matérn family of kernels, showing the NT kernels induce a very general class of models. We further discuss the implications of our analysis for some recent results on the regret bounds for reinforcement learning and bandit algorithms, which use overparameterized neural networks.

## 1 INTRODUCTION

Neural networks have shown an unprecedented success in the hands of practitioners in several fields. Examples of successful applications include classification (Krizhevsky et al., 2012; LeCun et al., 2015) and generative modeling (Goodfellow et al., 2014; Brown et al., 2020). This practical efficiency has accelerated developing analytical tools for understanding the properties of neural networks. A long standing dilemma in the analysis of neural networks is their favorable generalization error despite typically being extremely overparameterized. It is well known that the classical statistical learning methods for bounding the generalization error, such as Vapnik–Chervonenkis (VC) dimension (Bartlett, 1998; Anthony & Bartlett, 2009) and Rademacher complexity (Bartlett & Mendelson, 2002; Neyshabur et al., 2015), often result in vacuous bounds in the case of neural networks (Dziugaite & Roy, 2017).

Motivated with the problem mentioned above, we investigate the generalization error in overparameterized neural networks. In particular, let $\mathcal{D}_n = \{(x_i, y_i)\}_{i=1}^n$ be a possibly noisy dataset collected from a true model $f$: $y_i = f(x_i) + \epsilon_i$, where $\epsilon_i$ are well behaved zero mean noise terms. Let $\hat{f}_n$ be the learnt model using a gradient based training of an overparameterized neural network on $\mathcal{D}_n$. We are interested in bounding $|f(x) - \hat{f}_n(x)|$ uniformly at all test points $x$ belonging to a particular domain $\mathcal{X}$.

A recent line of work has shown that gradient based learning of certain overparameterized neural networks can be analyzed using a projection on the tangent space at random initialization. This leads to the equivalence of the solution of the neural network model to a kernel ridge regression, that is a linear regression in the possibly infinite dimensional feature space of the corresponding kernel. The particular kernel arising from this approach is called the neural tangent (NT) kernel (Jacot et al., 2018). This equivalence is based on the observation that, during training, the parameters of the neural network remain within a close vicinity of the random initialization. This regime is sometimes referred to as *lazy* training (Chizat et al., 2019).

## 1.1 CONTRIBUTIONS

Here, we summarize our contributions. We propose the maximal information gain (MIG), denoted by $\gamma_k(n)$, of the NT kernel $k$, as a measure of complexity of the learning problem for overparameterized neural networks in kernel regimes. The MIG is closely related to the effective dimension of the kernel (for the formal definition of the MIG and its connection to the effective dimension of the kernel, see Section 4). We provide kernel specific bounds on the MIG of the NT kernel. Using these bounds, for networks with infinite number of parameters, we analytically prove novel generalization error bounds decaying with respect to the dataset size $n$. Furthermore, our results capture the dependence of the decay rate of the error bound on the smoothness properties of the activation functions.

Our analysis crucially uses Mercer's theorem and the spectral decomposition of the NT kernel in the basis of spherical harmonics (which are special functions on the surface of the hypersphere; see Section 3). We derive bounds on the decay rate of the Mercer eigenvalues (referred to hereafter as *eigendecay*) based on the differentiability of the activation functions. These eigendecays are obtained by careful characterization of a recent result relating the eigendecay to the asymptotic endpoint expansions of the kernel, given in (Bietti & Bach, 2020). In particular, we consider $s - 1$ times differentiable activation functions $a_s(u) = (\max(0, u))^s, s \in \mathbb{N}$. We show that, with these activation functions and a $d$ dimensional hyperspherical support, the eigenvalues $\lambda_i$ of the corresponding NT kernel decay at the rate $\lambda_i \sim i^{-\frac{d+2s-2}{d-1}}$[1] (up to certain parity constraints on $i$ in special cases; see Section 2 for details).

We then use the eigendecay of the NT kernel to establish novel bounds on its MIG, which are of the form $\gamma_k(n) = \tilde{\mathcal{O}}(n^{\frac{d-1}{d+2s-2}})$. Then, we show this implies high probability error bounds of the form $\sup_{x \in \mathcal{X}} |f(x) - \hat{f}_n(x)| = \tilde{\mathcal{O}}(n^{\frac{-2s+1}{2d+4s-4}})$, when $f$ belongs to the corresponding reproducing kernel Hilbert space (RKHS), and $\mathcal{D}_n$ is distributed over $\mathcal{X}$ in a sufficiently informative way. In particular, we consider a $\mathcal{D}_n$ that is collected sequentially in a way that maximally reduces the uncertainty in the model. That results in a quasi-uniform dataset over the entire domain $\mathcal{X}$ (see Section 5.2 for details). A summary of our results on MIG and error bounds is reported in Table 1.

As a byproduct of our results, we show that the RKHS of the NT kernel with activation function $a_s(.)$ is equivalent to the RKHS of a Matérn kernel with smoothness parameter $\nu = s - \frac{1}{2}$. This result was already known for a special case. In particular, Geifman et al. (2020); Chen & Xu (2021) recently showed the equivalence between the RKHS of the NT kernel with ReLU activation function, $a_1(.)$, and that of the Matérn kernel with $\nu = \frac{1}{2}$ (also known as the Laplace kernel). Our contribution includes generalizing this result to the equivalence between the RKHS of the NT kernel under various activation functions and that of a Matérn kernel with the corresponding smoothness (see Section 3.2). We note that the members of the RKHS of Matérn family of kernels can uniformly approximate almost all continuous functions (Srinivas et al., 2010). Therefore, our results suggest that the NT kernels induce a very general class of models.

As an intermediate step, we also consider the simpler random feature (RF) model which is the result of partial training of the neural network. Also referred to as conjugate kernel or neural networks Gaussian process (NNGP) (Cho & Saul, 2009; Daniely et al., 2016; Lee et al., 2018; de G. Matthews et al., 2018), this model corresponds to the case where only the last layer of the neural network is trained (see Section 2 for details).

Table 1: The bounds on MIG and generalization error for RF and NT kernels, with $a_s(.)$ activation functions, when the true model $f$ belongs to the RKHS of the corresponding kernel and is supported on the hypersphere $\mathbb{S}^{d-1} \subset \mathbb{R}^d$.

| Kernel | MIG, $\gamma_k(n)$ | Error bound, $|f(x) - \hat{f}_n(x)|$ |
|---|---|---|
| NT | $\mathcal{O}\left(n^{\frac{d-1}{d+2s-2}}(\log(n))^{\frac{2s-1}{d+2s-2}}\right)$ | $\mathcal{O}\left(n^{\frac{-2s+1}{2d+4s-4}}(\log(n))^{\frac{d+4s-3}{2d+4s-4}}\right)$ |
| RF | $\mathcal{O}\left(n^{\frac{d-1}{d+2s}}(\log(n))^{\frac{2s+1}{d+2s}}\right)$ | $\mathcal{O}\left(n^{\frac{-2s-1}{2d+4s}}(\log(n))^{\frac{d+4s+1}{2d+4s}}\right)$ |

---

[1]The general notations used in this paper are defined in Appendix A.

We note that the decay rate of the error bound is faster in the case of RF kernel with the same activation function. Also, the decay rate of the error bound is faster when $s$ is larger, for both NT and RF kernels. In interpreting these results, we should however take into account the RKHS of the corresponding kernels. The RKHS of the RF kernel is smaller than the RKHS of the NT kernel. Also, when $s$ is larger, the RKHS is smaller for both NT and RF kernels.

In Section 6, we provide experimental results on the error rate for synthetic functions belonging to the RKHS of the NT kernel, which are consistent with the analytically predicted behaviors.

Our bounds on MIG may be of independent interest. In Section 7, we comment on the implications of our results for the regret bounds in reinforcement learning (RL) and bandit problems, which use overparameterized neural network models (e.g., Yang et al., 2020; Gu et al., 2021).

## 2 OVERPARAMETERIZED NEURAL NETWORKS IN KERNEL REGIMES

In this section, we briefly overview the equivalence of overparameterized neural network models to kernel methods. It has been recently shown that, in overparameterized regimes, gradient based training of a neural network reaches a global minimum, where the weights remain within a close vicinity of the random initialization (Jacot et al., 2018). In particular, let $f(x, W)$ denote a neural network with a large width $m$ parameterized by $W$. The model can be approximated with its linear projection on the tangent space at random initialization $W_0$, as

$$f(x, W) \approx f(x, W_0) + \langle W - W_0, \nabla_W f(x, W_0)\rangle.$$

The approximation error can be bounded by the second order term $\xi_m \|W - W_0\|^2$, and shown to be diminishing as $m$ grows, where $\xi_m$ is the spectral norm of the Hessian matrix of $f$ (Liu et al., 2020). The approximation becomes an equality when the width of the hidden layers approaches infinity. Known as lazy training regime (Chizat et al., 2019), this model is equivalent to the NT kernel (Jacot et al., 2018), given by

$$k(x, x') = \lim_{m \to \infty} \langle \nabla_W f(x, W_0), \nabla_W f(x', W_0)\rangle.$$

### 2.1 NEURAL KERNELS FOR A 2 LAYER NETWORK

Consider the following 2 layer neural network with width $m$ and a proper normalization

$$f(x, W) = \frac{c}{\sqrt{m}} \sum_{j=1}^{m} v_j a(w_j^\top x), \tag{1}$$

where $a(.)$ is the activation function, $c$ is a constant, $x \in \mathcal{X}$ is a $d$ dimensional input to the network, $w_j \in \mathbb{R}^d$ are the weights in the first layer, and $v_j \in \mathbb{R}$ are the weights in the second layer. The weights $w_j$ are initialized randomly according to $\mathcal{N}(\mathbf{0}_d, \boldsymbol{I}_d)$ and $v_j$ are initialized randomly according to $\mathcal{N}(0, 1)$. Then, the NT kernel of this network can be expressed as (Jacot et al., 2018)

$$k_{\text{NT}}(x, x') = c^2(x^\top x')\mathbb{E}_{w \sim \mathcal{N}(\mathbf{0}_d, \boldsymbol{I}_d)}[a'(w^\top x)a'(w^\top x')] + c^2\mathbb{E}_{w \sim \mathcal{N}(\mathbf{0}_d, \boldsymbol{I}_d)}[a(w^\top x)a(w^\top x')]. \tag{2}$$

When $w_j$ are fixed at initialization, and only $v_j$ are trained with $\ell^2$ regularization, the model is equivalent to a certain Gaussian process (GP) (Neal, 2012) that is often referred to as a random feature (RF) model. The RF kernel is given as (Rahimi et al., 2007)

$$k(x, x') = c^2\mathbb{E}_{w \sim \mathcal{N}(\mathbf{0}_d, \boldsymbol{I}_d)}[a(w_j^\top x)a(w^\top x')]. \tag{3}$$

Note that the RF kernel is the same as the second term in the expression of the NT kernel given in equation 2.

### 2.2 ROTATIONAL INVARIANCE

When the input domain $\mathcal{X}$ is the hypersphere $\mathbb{S}^{d-1}$, by the spherical symmetry of the Gaussian distribution, the neural kernels are invariant to unitary transformations[2] and can be expressed as

---

[2]Rotationally invariant kernels are also sometimes referred to as *dot product* or *zonal* kernels.

a function of $x^\top x'$. We thus use the notations $\kappa_{\mathrm{NT},s}(x^\top x')$ and $\kappa_s(x^\top x')$ to refer to the NT and RF kernels, as defined in equation 2 and equation 3, respectively. In this notation, $s$ specifies the smoothness of the activation function, $a_s(u) = (\max(0, u))^s$.

The particular form of the neural kernels depends on $s$. For example, for the special case of $s = 1$, which corresponds to the ReLU activation function, the following closed form expressions can be derived from the expressions given in equation 2 and equation 3.

$$
\begin{aligned}
\kappa_{\mathrm{NT},1}(u) &= \frac{u}{\pi}(\pi - \arccos(u)) + \kappa_1(u), \\
\kappa_1(u) &= \frac{1}{\pi}\left(u(\pi - \arccos(u)) + \sqrt{1 - u^2}\right).
\end{aligned}
$$

In deriving the expressions above, the constant $c$ in equations 2 and 3 is set to $\sqrt{2}$. In general, we choose $c^2 = \frac{2}{(2s-1)!!}$, that normalizes the maximum value of the RF kernel to 1: $\kappa_s(1) = 1$, $\forall s \geq 1$.

## 2.3 NEURAL KERNELS FOR MULTILAYER NETWORKS

For an $l > 2$ layer network, the NT kernel is recursively given as (see, Jacot et al., 2018, Theorem 1)

$$
\begin{aligned}
\kappa_{\mathrm{NT},s}^l(u) &= c^2 \kappa_{\mathrm{NT},s}^{l-1}(u) \kappa_s'(\kappa^{l-1}(u)) + \kappa_s^l(u), \\
\kappa_s^l(u) &= \kappa_s(\kappa_s^{l-1}(u)).
\end{aligned} \tag{4}
$$

Here, $\kappa_s^l(.)$ corresponds to the RF kernel of an $l$ layer network, as shown in Cho & Saul (2009); Daniely et al. (2016); Lee et al. (2018); de G. Matthews et al. (2018). In our notation, when $l = 2$, we drop the layer index $l$ from the superscript, as in the previous subsection. An expression for the derivative $\kappa_s'(.)$ of the RF kernel in a 2 layer network, based on $\kappa_{s-1}$, is given in Lemma 1. That is needed to see the equivalence of the expressions in equation 4 to the one given in Theorem 1 of Jacot et al. (2018).

## 3 THE RKHS OF THE NEURAL KERNELS

When the domain is $\mathbb{S}^{d-1}$, as discussed earlier, the neural kernels have a rotationally invariant (dot product) form. The rotationally invariant kernels can be diagonalized in the basis of spherical harmonics, which are special functions on the surface of the hypersphere. Specifically, the Mercer decomposition of the kernels (for all $l \geq 2$ and $s \geq 1$) can be given as

$$
\kappa(x^\top x') = \sum_{i=1}^{\infty} \sum_{j=1}^{N_{d,i}} \tilde{\lambda}_i \tilde{\phi}_{i,j}(x) \tilde{\phi}_{i,j}(x'),
$$

where the eigenfunction $\tilde{\phi}_{i,j}$ is the $j$'th spherical harmonic of degree $i$, $\tilde{\lambda}_i$ is the corresponding eigenvalue, $N_{d,i} = \frac{2i+d-2}{i}\binom{i+d-3}{d-2}$ is the multiplicity of $\tilde{\lambda}_i$ (meaning the number of spherical harmonics of degree $i$, which all share the same eigenvalue $\tilde{\lambda}_i$), and the conver-

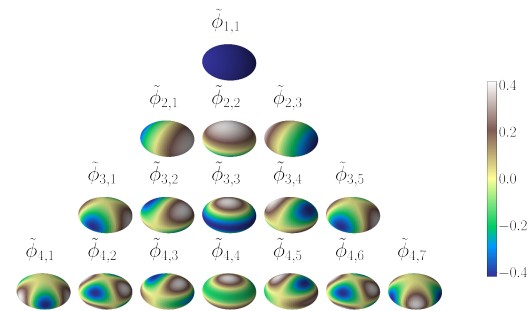

Figure 1: The spherical harmonics of degrees $i = 1, 2, 3, 4$, on $\mathbb{S}^2$. The function value is given by color.

gence of the series holds uniformly on $\mathcal{X} \times \mathcal{X}$. As a consequence of Mercer's representation theorem, the corresponding RKHS can be written as

$$
\mathcal{H}_\kappa = \left\{ f(\cdot) = \sum_{i=1}^{\infty} \sum_{j=1}^{N_{d,i}} w_{i,j} \tilde{\lambda}_i^{\frac{1}{2}} \tilde{\phi}_{i,j}(\cdot) : w_{i,j} \in \mathbb{R}, \|f\|_{\mathcal{H}_\kappa}^2 = \sum_{i=1}^{\infty} \sum_{j=1}^{N_{d,i}} w_{i,j}^2 < \infty \right\}.
$$

Formal statements of Mercer's theorem and Mercer's representation theorem, as well as a formal definition of Mercer eigenvalues and eigenfunctions are given in Appendix B.

We use the notation $\lambda_i$ (in contrast to $\tilde{\lambda}_i$) to denote the decreasingly ordered eigenvalues, when their multiplicity is taken into account (for all $i : \sum_{i''=1}^{i'-1} N_{d,i''} < i \leq \sum_{i''=1}^{i'} N_{d,i''}$, we define $\lambda_i = \tilde{\lambda}_{i'}$).

### 3.1 THE DECAY RATE OF THE EIGENVALUES

As discussed, both RF and NT kernels can be represented in the basis of spherical harmonics. Our analysis of the MIG relies on the decay rate of the corresponding eigenvalues. In this section, we derive the eigendecay depending on the particular activation functions.

In the special case of ReLU activation function, several recent works (Geifman et al., 2020; Chen & Xu, 2021; Bietti & Bach, 2020) have proven a $\tilde{\lambda}_i \sim i^{-d}$ eigendecay for the NT kernel on the hypersphere $\mathbb{S}^{d-1}$. In this work, we use a recent result from Bietti & Bach (2020) to derive the eigendecay based on the differentiability properties of the activation functions.

**Proposition 1** *Consider the neural kernels on the hypersphere $\mathbb{S}^{d-1}$, and their Mercer decomposition in the basis of spherical harmonics. Then, the eigendecays are as reported in Table 2.*

Table 2: The eigendecays of the neural kernels.

| | | |
|---|---|---|
| NT | $l = 2$ | For $i$ of the same parity of $s$: $\tilde{\lambda}_i \sim i^{-d-2s+2}$, $\lambda_i \sim i^{-\frac{d+2s-2}{d-1}}$ 
 For $i$ of the opposite parity of $s$: $\tilde{\lambda}_i = o(i^{-d-2s+2})$, $\lambda_i = o(i^{-\frac{d+2s-2}{d-1}})$ |
| | $l > 2$ | $\tilde{\lambda}_i \sim i^{-d-2s+2}$, $\lambda_i \sim i^{-\frac{d+2s-2}{d-1}}$ |
| RF | $l = 2$ | For $i$ of the opposite parity of $s$: $\tilde{\lambda}_i \sim i^{-d-2s}$, $\lambda_i \sim i^{-\frac{d+2s}{d-1}}$ 
 For $i$ of the same parity of $s$: $\tilde{\lambda}_i = o(i^{-d-2s})$, $\lambda_i = o(i^{-\frac{d+2s}{d-1}})$ |
| | $l > 2$ | $\tilde{\lambda}_i \sim i^{-d-2s}$, $\lambda_i \sim i^{-\frac{d+2s}{d-1}}$ |

*Proof Sketch.* The proof follows from Theorem 1 of Bietti & Bach (2020), which proved that the eigendecay of a rotationally invariant kernel $\kappa(.) : [-1, 1] \to \mathbb{R}$ can be determined based on its asymptotic expansions around the endpoints, $\pm 1$. In particular, when

$$\kappa(1 - t) = p_{+1}(t) + c_{+1}t^\theta + o(t^\theta),$$
$$\kappa(-1 + t) = p_{-1}(t) + c_{-1}t^\theta + o(t^\theta),$$

for polynomial $p_{\pm 1}$ and non-integer $\theta$, the eigenvalues decay at the rate $\tilde{\lambda}_i \sim (c_{+1} + c_{-1})i^{-d-2\theta+1}$, for even $i$, and $\tilde{\lambda}_i \sim (c_{+1} - c_{-1})i^{-d-2\theta+1}$, for odd $i$. A formal statement of this result is given in Appendix E. They derived these expansions for the case of the ReLU activation function. For completeness, we derive the asymptotic endpoint expansions for the neural kernels with $s - 1$ times differentiable activation functions, in the case of $l = 2$ and $l > 2$ layer networks. Our derivations is based on the following Lemma.

**Lemma 1** *For the RF kernel defined in equation 3, with activation function $a_s(.)$, and $c = \frac{2}{(2s-1)!!}$, we have*

$$\kappa'_s(u) = \frac{s^2}{2s-1}\kappa_{s-1}(u).$$

The proof is given in Appendix D. Lemma 1 allows us to recursively derive the endpoint expansions of $\kappa_s$ from those of $\kappa_{s-1}$, through integration. We thus obtain $\theta$ and $c_{\pm 1}$ for $s > 1$, when $l = 2$, using a recursive relation over $s$. We then use the compositional forms given in equation 4 to obtain the same, when $l > 2$, resulting in the eigendecays reported in Table 2. $\square$

**Remark 1** *The parity constraints in Table 2 may imply undesirable behavior for 2 layer infinitely wide networks. Ronen et al. (2019) studied this behavior and showed that adding a bias term removes these constraints, for the NT kernel. Although the parity constrained eigendecay affects the*

*representation power of the neural kernels, it does not affect our analysis of MIG and error rates, since for that analysis we only use $\tilde{\lambda}_i = \mathcal{O}(i^{-d-2s+2})$.*

**Remark 2** *We note that our results do not necessarily apply to deep neural networks in the sense that the implied constants in Table 2 grow exponentially in $l$, for $s > 1$ (see Appendix E for the expressions of the constants). When $s = 1$, however, the constants grow at most as fast as $l^2$ (as shown in Bietti & Bach, 2020).*

### 3.2 EQUIVALENCE OF THE RKHSs OF THE NEURAL AND MATÉRN KERNELS

Our bounds on the eigendecay of the neural kernels show the connection between the RKHS of the neural kernels, and that of Metérn family of kernels. Let $\mathcal{H}_{k_\nu}$ denote the RKHS of the Matérn kernel with smoothness parameter $\nu$.

**Theorem 1** *On a hyperspherical domain, when $l > 2$, $\mathcal{H}_{\kappa_{NT,s}^l}$ ($\mathcal{H}_{\kappa_s^l}$) is equivalent to $\mathcal{H}_{k_\nu}$, and, when $l = 2$, $\mathcal{H}_{\kappa_{NT,s}^l} \subset \mathcal{H}_{k_\nu}$ ($\mathcal{H}_{\kappa_s^l} \subset \mathcal{H}_{k_\nu}$), where $\nu = s - \frac{1}{2}$ ($\nu = s + \frac{1}{2}$). The statement is summarized in Table 3.*

Table 3: The relation between the RKHSs corresponding to the neural and Matérn kernels.

| Kernel | $l = 2$ | $l > 2$ |
|---|---|---|
| NT | $\mathcal{H}_{\kappa_{NT,s}} \subset \mathcal{H}_{k_{s-\frac{1}{2}}}$ | $\mathcal{H}_{\kappa_{NT,s}} \equiv \mathcal{H}_{k_{s-\frac{1}{2}}}$ |
| RF | $\mathcal{H}_{\kappa_s} \subset \mathcal{H}_{k_{s+\frac{1}{2}}}$ | $\mathcal{H}_{\kappa_s} \equiv \mathcal{H}_{k_{s+\frac{1}{2}}}$ |

The formal definition of equivalence ($\equiv$) and subset ($\subset$) relation between two normed spaces is given in Appendix A.

*Proof Sketch.* The proof follows from the eigendecay of the neural kernels given in Table 2, the eigendecay of the Matérn family of kernels on manifolds proven in Borovitskiy et al. (2020), and the Mercer's representation theorem. Details are given in Appendix F.

$\square$

**Remark 3** *The observations that when $l = 2$, we only have $\mathcal{H}_{\kappa_{NT,s}^l} \subset \mathcal{H}_{k_{s-\frac{1}{2}}}$, and not vice versa, is a result of parity constraints in Table 2. The subset relation can change to equivalence, with a bias term which removes the parity constraints (see, Ronen et al., 2019).*

A special case of Theorem 1, when $s = 1$, was recently proven in (Geifman et al., 2020; Chen & Xu, 2021). This special case pertains to the ReLU activation function and the Matérn kernel with smoothness parameter $\nu = \frac{1}{2}$, that is also referred to as the Laplace kernel.

**Remark 4** *It is known that the RKHS of a Matérn kernel with smoothness parameter $\nu$ is equivalent to a Sobolev space with parameter $\nu + \frac{d}{2}$ (e.g., see, Kanagawa et al., 2018, Section 4). This observation provides an intuitive interpretation for the RKHS of the neural kernels as a consequence of Theorem 1. That is $\|f\|_{\kappa_{NT,s}^l}$ ($\|f\|_{\kappa_s^l}$) is proportional to the cumulative $L^2$ norm of the weak derivatives of $f$ up to $s + \frac{d-1}{2}$ ($s + \frac{d+1}{2}$) order. I.e., $f \in \mathcal{H}_{\kappa_{NT,s}^l}$ ($f \in \mathcal{H}_{\kappa_s^l}$) translates to the existence of weak derivatives of $f$ up to $s + \frac{d-1}{2}$ ($s + \frac{d+1}{2}$) order, which can be understood as a versatile measure for the smoothness of $f$ controlled by $s$. For the details on the definition of weak derivatives and Sobolev spaces see, e.g., Hunter & Nachtergaele (2011).*

## 4 MAXIMAL INFORMATION GAIN OF THE NEURAL KERNELS

In this section, we use the eigendecay of the neural kernels shown in the previous section to derive bounds on their information gain, which will then be used to prove uniform bounds on the generalization error.

## 4.1 INFORMATION GAIN: A MEASURE OF COMPLEXITY

To define information gain, we need to introduce a fictitious GP model. Assume $F$ is a zero mean GP indexed on $\mathcal{X}$ with kernel $k$. Information gain then refers to the mutual information $\mathcal{I}(Y_n; F)$ between the data values $Y_n = [y_i]_{i=1}^n$ and $F$. From the closed from expression of mutual information between two multivariate Gaussian distributions (Cover, 1999), it follows that

$$\mathcal{I}(Y_n; F) = \frac{1}{2} \log \det \left( \boldsymbol{I}_n + \frac{1}{\lambda^2} \mathbf{K}_n \right).$$

Here $\mathbf{K}_n$ is the kernel matrix $[\mathbf{K}_n]_{i,j} = k(x_i, x_j)$, and $\lambda > 0$ is a regularization parameter. We also define the data independent and kernel specific maximal information gain (MIG) as follows.

$$\gamma_k(n) = \sup_{X_n \subset \mathcal{X}} \mathcal{I}(Y_n; F). \tag{5}$$

The MIG is also closely related to the *effective dimension* of the kernel. In particular, although the feature space of the neural kernels are infinite dimensional, for a finite dataset, the number of features with a significant impact on the regression model can be finite. The effective dimension of a kernel is often defined as (Zhang, 2005; Valko et al., 2013)

$$\tilde{d}_k(n) = \mathrm{Tr} \left( \mathbf{K}_n (\mathbf{K}_n + \lambda^2 \mathbf{I}_n)^{-1} \right). \tag{6}$$

It is known that the information gain and the effective dimension are the same up to logarithmic factors. Specifically, $\tilde{d}_k(n) \le \mathcal{I}(Y_n; F)$, and $\mathcal{I}(Y_n; F) = \mathcal{O}(\tilde{d}_k(n) \log(n))$ (Calandriello et al., 2019).

Relating the information gain to the effective dimension provides some intuition into why it can capture the complexity of the learning problem. Roughly speaking, our bounds on the generalization error are analogous to the ones for a linear model, which has a feature dimension of $\tilde{d}_k(n)$.

## 4.2 BOUNDING THE INFORMATION GAIN OF THE NEURAL KERNELS

In the following theorem, we provide a bound on the maximal information gain of the neural kernels.

**Theorem 2** *For the neural kernels with all $l \ge 2$, on a hyperspherical domain $\mathcal{X} = \mathbb{S}^{d-1}$, we have*

$$\gamma_{\kappa_{NT,s}^l}(n) = \mathcal{O} \left( n^{\frac{d-1}{d+2s-2}} (\log(n))^{\frac{2s-1}{d+2s-2}} \right), \quad \text{in the case of NT kernel,}$$

$$\gamma_{\kappa_s^l}(n) = \mathcal{O} \left( n^{\frac{d-1}{d+2s}} (\log(n))^{\frac{2s+1}{d+2s}} \right), \quad \text{in the case of RF kernel.}$$

A special case of our Theorem 2, for the special case of the ReLU activation function, was recently proved in Kassraie & Krause (2021).

*Proof Sketch.* The main components of the analysis include the eigendecay given in Proposition 1, a projection on a finite dimensional RKHS technique proposed in Vakili et al. (2021b), and a bound on the sum of spherical harmonics of degree $i$ determined by the Legendre polynomials, that is sometimes referred to as the Legendre *addition* theorem (Maleček & Nádeník, 2001). Details are given in Appendix G.

□

# 5 UNIFORM BOUNDS ON THE GENERALIZATION ERROR

In this section, we use the bound on MIG from previous section to establish uniform bounds on the generalization error.

## 5.1 IMPLICIT ERROR BOUNDS

We first overview the implicit (data dependent) error bounds for the kernel methods under the following assumptions.

**Assumption 1** *Assume the true data generating model $f$ is in the RKHS corresponding to a neural kernel $\kappa$. In particular, $\|f\|_{\mathcal{H}_\kappa} \leq B$, for some $B > 0$. In addition, assume that the observation noise $\epsilon_i$ are independent $R$ sub-Gaussian random variables. That is $\mathbb{E}[\exp(\eta \epsilon_i)] \leq \exp(\frac{\eta^2 R^2}{2})$, $\forall \eta \in \mathbb{R}, \forall i \geq 1$, where $y_i = f(x_i) + \epsilon_i, \forall i \geq 1$.*

Under Assumption 1, for a fixed $x \in \mathcal{X}$, we have, with probability at least $1-\delta$ (Vakili et al., 2021a),

$$|f(x) - \hat{f}_n(x)| \leq \beta(\delta)\sigma_n(x), \tag{7}$$

where $\hat{f}_n(x) = \mathbf{k}_n^\top(x)(\mathbf{K}_n + \lambda^2 \mathbf{I}_n)^{-1} Y_n$ is the solution to the kernel ridge regression using $\kappa$ and $\mathcal{D}_n$, $\sigma_n^2(x) = k(x,x) - \mathbf{k}_n^\top(x)(\mathbf{K}_n + \lambda^2 \mathbf{I}_n)^{-1} \mathbf{k}_n(x)$, $\mathbf{k}_n^\top(x) = [\kappa(x^\top x_i)]_{i=1}^n$, and $\beta(\delta) = B + \frac{R}{\lambda}\sqrt{2\log(\frac{2}{\delta})}$. Equation 7 provides a high probability bound on $|f(x) - \hat{f}_n(x)|$. The decay rate of this bound based on $n$, however, is not explicit.

## 5.2 Explicit Error Bounds

In this section, we use the bounds on MIG and the implicit error bounds from the previous subsection to provide explicit (in $n$) bounds on the error. It is clear that with no assumption on the distribution of the data, generalization error cannot be nontrivially bounded. For example, if all the data points are collected from a small region of the input domain, it is not expected for the error to be small far from this small region. We here consider a dataset that is distributed over the input space in a sufficiently informative way. For this purpose, we introduce a data collection module which collects the data points based on the current uncertainty level of the kernel model. In particular, consider a dataset $\tilde{\mathcal{D}}_n$ collected as follows: $x_i = \arg\max_{x \in \mathcal{X}} \sigma_{i-1}(x)$, where $\sigma_i(.)$ is defined above. We have the following result.

**Theorem 3** *Consider the neural kernels with $l \geq 2$, on a hyperspherical domain $\mathcal{X} = \mathbb{S}^{d-1}$. Consider a model $\hat{f}_n$ trained on $\tilde{\mathcal{D}}_n$. Under Assumption 1, with probability at least $1 - \delta$, uniformly in $x \in \mathcal{X}$,*

$$|f(x) - \hat{f}_n(x)| = \mathcal{O}\left(n^{\frac{-2s+1}{2d+4s-4}}(\log(n))^{\frac{2s-1}{2d+4s-4}}(\log(\frac{n^{d-1}}{\delta}))^{\frac{1}{2}}\right), \quad \text{in the case of NT kernel,}$$

$$|f(x) - \hat{f}_n(x)| = \mathcal{O}\left(n^{\frac{-2s-1}{2d+4s}}(\log(n))^{\frac{2s+1}{2d+4s}}(\log(\frac{n^{d-1}}{\delta}))^{\frac{1}{2}}\right), \quad \text{in the case of RF kernel.} \tag{8}$$

*Proof Sketch.* The proof follows the same steps as in the proof of Theorem 3 of Vakili et al. (2021a). The key step is bounding the total uncertainty in the model with the information gain that is $\sum_{i=1}^n \sigma_{i-1}^2(x_i) \leq \frac{2}{\log(1+\frac{1}{\lambda^2})}\mathcal{I}(Y_n; F)$ (Srinivas et al., 2010). This allows us to bound the $\sigma_n$ in the right hand side of equation 7 by $\sqrt{\gamma_\kappa(n)/n}$ up to multiplicative absolute constants. Plugging in the bounds on $\gamma_\kappa(n)$ from Theorem 2, and using the implicit error bound given in equation 7 (with a union bound on a discretization of the domain with size $\mathcal{O}(n^{d-1})$), we obtain equation 8. A detailed proof is given in Appendix H.

$\square$

## 6 Experiments

In this section, we provide experimental results on the error rates. In our experiments, we create synthetic data from a true model $f$ that belongs to the RKHS of a NT kernel $\kappa$. For this purpose, we create two random vectors $\hat{X}_{n_0} \in \mathcal{X}^{n_0}$ and $\hat{Y}_{n_0} \in \mathbb{R}^{n_0}$, and let $f(.) = \hat{\mathbf{k}}_{n_0}^\top(.)(\hat{\mathbf{K}}_{n_0} + \delta^2 \mathbf{I}_{n_0})^{-1}\hat{Y}_{n_0}$, where $\hat{\mathbf{k}}_{n_0}$ and $\hat{\mathbf{K}}_{n_0}$ are defined similar to $\mathbf{k}_n$ and $\mathbf{K}_n$ in Subsection 5.1. We then generate datasets $\mathcal{D}_n$ of various sizes $n = 2^i$, with $i = 1, 2, \ldots, 13$, according to the underlying model $f$. We train the model to obtain $\hat{f}_n$. Figure 2 shows the error rate versus the size $n$ of the dataset for various $d$ and $s$. The experiments show that the error converges to zero at a rate satisfying the bounds given in Theorem 3. In addition, as analytically predicted, the absolute value of the error rate exponent increases with $s$ and decreases with $d$. Our experiments use the *neural-tangents* library (Novak et al., 2019). See Appendix I for more details.

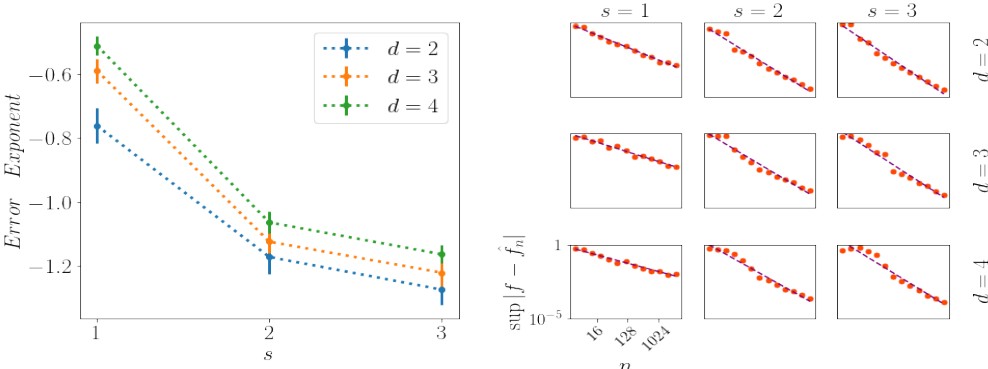

Figure 2: Left: the exponent of the error rate versus $s$ and $d$. As expected, larger values of $s$ and $d$ result in, respectively, faster and slower error decays. The bars show the standard deviation. Right: error rates for $s = 1, 2, 3$ and $d = 2, 3, 4$, shown in separate panels. Both axes are in log scale so that the slope of the line represents the exponent of the error rate. Experimental error rates are consistent with the analytically predicted results.

## 7 DISCUSSION

Our bounds on MIG may be of independent interest in other problems. Recent works on RL and bandit problems, which use overparameterized neural network models, derived an $\tilde{\mathcal{O}}(\gamma_k(n)\sqrt{n})$ regret bound (see, e.g., Zhou et al., 2020; Yang et al., 2020; ZHANG et al., 2021; Gu et al., 2021). These works however did not characterize the bound on $\gamma_k(n)$, leaving the regret bounds implicit. A consequence of our bounds on $\gamma_k(n)$ is an explicit (in $n$) regret bound for these RL and bandit problems. Our results however have mixed implications by showing that the existing regret bounds are not necessarily sublinear in $n$. In particular, plugging in our bound on $\gamma_{\kappa_{\mathrm{NT},s}}(n)$ into the existing regret bounds, we get $\tilde{\mathcal{O}}(n^{\frac{1.5d+s-2}{d+2s-2}})$, which is sublinear only when $s > \frac{d}{2}$ (that, e.g., excludes the ReLU activation function). Our analysis of various activation functions is thus essential in showing sublinear regret bounds for at least some cases. As recently formalized in Vakili et al. (2021c), it remains an open problem whether the analysis of RL and bandit algorithms in kernel regimes can improve to have an always sublinear regret bound. We note that this open problem and the analysis of RL and bandit problems were not considered in this work. Our remark is mainly concerned with the consequences of our bound on MIG, which appears in the regret bounds.

The recent work of Wang et al. (2020) proved that the $L^2$ norm of the error is bounded as $\|f - \hat{f}_n\|_{L^2} = \mathcal{O}(n^{-\frac{d}{2d-1}})$, for a two layer neural network with ReLU activation functions in kernel regimes. Bordelon et al. (2020) decomposed the $L^2$ norm of the error into a series corresponding to eigenfunctions of the NT kernel and provided bounds based on the corresponding eigenvalues. Our results are stronger as they are given in terms of absolute error instead of $L^2$ norm. In addition, we provide explicit error bounds depending on differentiability of the activation functions.

The MIG has been studied for popular GP kernels such as Matérn and Squared Exponential (Srinivas et al., 2010; Vakili et al., 2021b). Our proof technique is similar to that of Vakili et al. (2021b). Their analysis however does not directly apply to the NT kernel on the hypersphere. The reason is that Vakili et al. (2021b) assume uniformly bounded eigenfunctions. In our analysis, we use a Mercer decomposition of the NT kernel in the basis of spherical harmonics. Those are not uniformly bounded. Technical details are provided in the analysis of Theorem 2. Applying a proof technique similar to Srinivas et al. (2010) results in suboptimal bounds in our case.

Our work may be relevant to the recent developments in the field of *implicit neural representations* (Mildenhall et al., 2020; Sitzmann et al., 2020; Fathony et al., 2020). In this line of work, the input to a neural network often represents the coordinates of a plane (image), a camera position or angle, and the function to be approximated is the image or the scene itself. Interestingly, it has been observed that smooth activation functions perform better than ReLU in these applications, since the models are also smooth (Sitzmann et al., 2020).

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

## A  GENERAL NOTATION

In this section, we formally define our general notations. We use the notations $\mathbf{0}_n$ and $\boldsymbol{I}_n$ to denote the zero vector and the square identity matrix of dimension $n$, respectively. For a matrix $M$ (a vector $v$), $M^\top$ ($v^\top$) denotes its transpose. In addition, $\det$ and $\log \det$ are used to denote determinant of $M$ and its logarithm, respectively. The notation $\|v\|_{l^2}$ is used to denote the $l^2$ norm of a vector $v$. For $v \in \mathbb{R}^d$ and a symmetric positive definite matrix $\Sigma \in \mathbb{R}^{d \times d}$, $\mathcal{N}(v, \Sigma)$ denotes a normal distribution with mean $v$ and covariance $\Sigma$. The Kronecker delta is denoted by $\delta_{i,j}$. The notation $\mathbb{S}^{d-1}$ denotes the $d$ dimensional hypersphere in $\mathbb{R}^d$. For example, $\mathbb{S}^2 \subset \mathbb{R}^3$ is the usual sphere. The notations $o$ and $\mathcal{O}$ denote the standard mathematical orders, while $\tilde{\mathcal{O}}$ is used to denote $\mathcal{O}$ up to logarithmic factors. For two sequences $a_n, b_n : \mathbb{N} \to \mathbb{R}$, we use the notation $a_n \sim b_n$, when $a_n = \mathcal{O}(b_n)$ and $b_n = \mathcal{O}(a_n)$. For $s \in \mathbb{N}$, we define $(2s-1)!! = \prod_{i=1}^s (2i-1)$. For example, $3!! = 3$ and $5!! = 15$.

For a normed space $\mathcal{H}$, we use $\|.\|_{\mathcal{H}}$ to denote the norm associated with $\mathcal{H}$. For two normed spaces $\mathcal{H}_1, \mathcal{H}_2$, we write $\mathcal{H}_1 \subset \mathcal{H}_2$, if the following two conditions are satisfied. First, $\mathcal{H}_1 \subset \mathcal{H}_2$ as sets. Second, there exist a constant $c_1 > 0$, such that $\|f\|_{\mathcal{H}_2} \le c_1 \|f\|_{\mathcal{H}_1}$, for all $f \in \mathcal{H}_1$. We also write $\mathcal{H}_1 \equiv \mathcal{H}_2$, when both $\mathcal{H}_1 \subset \mathcal{H}_2$ and $\mathcal{H}_2 \subset \mathcal{H}_1$.

The derivative of $a : \mathbb{R} \to \mathbb{R}$ is denoted by $a'$. We define $0^0 = 0$, so that $a_0(x) = (\max(0, x))^0$ corresponds to the step function: $a_0(x) = 0$, when $x \le 0$, and $a_0(x) = 1$, when $x > 0$.

## B  MERCER'S THEOREM

In this section, we give an overview of Mercer's theorem, as well as the the reproducing kernel Hilbert spaces (RKHSs) associated with the kernels. Mercer's theorem (Mercer, 1909) provides a spectral decomposition of the kernel in terms of an infinite dimensional feature map (see, e.g. Steinwart & Christmann (2008), Theorem 4.49).

**Theorem 4 (Mercer's Theorem )** *Let $\mathcal{X}$ be a compact domain. Let $k$ be a continuous square integrable kernel with respect to a finite Borel measure $\mu$. Define a positive definite operator $T_k$*

$$(T_k f)(.) = \int_{\mathcal{X}} k(., x) f(x) d\mu.$$

*Then, there exists a sequence of eigenvalue-eigenfunction pairs $\{(\lambda_i, \phi_i)\}_{i=1}^\infty$ such that $\lambda_i \in \mathbb{R}^+$, and $T_k \phi_i = \lambda_i \phi_i$, for $i \ge 1$. Moreover, the kernel function can be represented as*

$$k(x, x') = \sum_{i=1}^\infty \lambda_i \phi_i(x) \phi_i(x'),$$

*where the convergence of the series holds uniformly on $\mathcal{X} \times \mathcal{X}$.*

The $\lambda_i$ and $\phi_i$ defined in Mercer's theorem are referred to as Mercer eigenvalues and Mercer eigenfunctions, respectively.

Let $\mathcal{H}_k$ denote the RKHS corresponding to $k$, defined as a Hilbert space equipped with an inner product $\langle ., . \rangle_{\mathcal{H}_k}$ satisfying the following: $k(., x) \in \mathcal{H}_k, \forall x \in \mathcal{X}$, and $\langle f, k(., x) \rangle_{\mathcal{H}_k} = f(x)$, $\forall x \in \mathcal{X}, \forall f \in \mathcal{H}_k$ (reproducing property). As a consequence of Mercer's theorem, $\mathcal{H}_k$ can be represented in terms of $\{(\lambda_i, \phi_i)\}_{i=1}^\infty$, that is often referred to as Mercer's representation theorem (see, e.g., Steinwart & Christmann (2008), Theorem 4.51).

**Theorem 5 (Mercer's Representation Theorem)** *Let $\{(\lambda_i, \phi_i)\}_{i=1}^\infty$ be the Mercer eigenvalue-eigenfunction pairs. Then, the RKHS corresponding to $k$ is given by*

$$\mathcal{H}_k = \left\{ f(\cdot) = \sum_{i=1}^\infty w_i \lambda_i^{\frac{1}{2}} \phi_i(\cdot) : w_i \in \mathbb{R}, \|f\|_{\mathcal{H}_k}^2 := \sum_{i=1}^\infty w_i^2 < \infty \right\}.$$

Mercer's representation theorem indicates that $\{\lambda_i^{\frac{1}{2}} \phi_i\}_{i=1}^\infty$ form an orthonormal basis for $\mathcal{H}_k$: $\langle \lambda_i^{\frac{1}{2}} \phi_i, \lambda_{i'}^{\frac{1}{2}} \phi_{i'} \rangle_{\mathcal{H}_k} = \delta_{i,i'}$. It also provides a constructive definition for the RKHS as the span of this orthonormal basis, and a constructive definition for the $\|f\|_{\mathcal{H}_k}$ as the $l^2$ norm of the weights $[w_i]_{i=1}^\infty$ vector.

## C CLASSICAL APPROACHES TO BOUNDING THE GENERALIZATION ERROR

There are classical statistical learning methods which address the generalization error in machine learning models. Two notable approaches are VC dimension (Bartlett, 1998; Anthony & Bartlett, 2009) and Rademacher complexity (Bartlett & Mendelson, 2002; Neyshabur et al., 2015). In the former, the expected generalization loss is bounded by the square root of the VC dimension divided by the square root of $n$. In the latter, the expected generalization loss scales with the Rademacher complexity of the hypothesis class. In the case of a 2 layer neural network of width $m$, it is shown that both of these approaches result in a generalization bound of $\mathcal{O}(\sqrt{m/n})$ that is vacuous for overparameterized neural networks with a very large $m$.

A more recent approach to studying the generalization error is PAC Bayes. It provides non-vacuous error bounds that depend on the distribution of parameters (Dziugaite & Roy, 2017). However, the distribution of parameters must be known or be estimated in order to use those bounds. Taking a different approach, Arora et al. (2019) provides bounds on expected generalization error of well-conditioned 2 layer neural networks based on the Gram matrix of the input data, which does not depend on the distribution of model parameters as in PAC Bayes. In contrast to all of these results, our bounds are explicit and are much stronger in the sense that they hold uniformly in $x$.

## D PROOF OF LEMMA 1

Lemma 1 offers a recursive relation over $s$ for the RF kernel. We prove the lemma by taking the derivative of $\kappa_s(.)$, and applying the Stein's lemma (given at the end of this section).

Let $x = [0, 0, \ldots, 0, 1]^\top$ and $x' = [0, 0, \ldots, \sqrt{1-u^2}, u]^\top$, so that $x^\top x' = u$, and $x, x' \in \mathbb{S}^{d-1}$. We have

$$
\frac{\partial}{\partial u} \mathbb{E}_{w \sim \mathcal{N}(\mathbf{0}_d, \mathbf{I}_d)} \left[ a_s(w^\top x) a_s(w^\top x') \right] = \mathbb{E}_{w \sim \mathcal{N}(\mathbf{0}_d, \mathbf{I}_d)} \left[ a_s(w^\top x) s a_{s-1}(w^\top x')(w_d - \frac{u w_{d-1}}{\sqrt{1-u^2}}) \right]
$$

$$
= s \mathbb{E}_{w \sim \mathcal{N}(\mathbf{0}_d, \mathbf{I}_d)} \Big[ s a_{s-1}(w^\top x) a_{s-1}(w^\top x')
$$

$$
+ (s-1) u a_s(w^\top x) a_{s-2}(w^\top x') \Big] - s \mathbb{E}_{w \sim \mathcal{N}(\mathbf{0}_d, \mathbf{I}_d)} \left[ \frac{(s-1)u}{\sqrt{1-u^2}} \sqrt{1-u^2} a_s(w^\top x) a_{s-2}(w^\top x') \right]
$$

$$
= s^2 \mathbb{E}_{w \sim \mathcal{N}(\mathbf{0}_d, \mathbf{I}_d)} \left[ a_{s-1}(w^\top x) a_{s-1}(w^\top x') \right].
$$

The first equation is obtained by taking derivative of the term inside the expectation. The second equation is obtained by applying Stein's lemma to $\mathbb{E}[a_s(w^\top x) s a_{s-1}(w^\top x') w_d]$, where $w_d$ is the normally distributed random variable, also to $\mathbb{E}[a_s(w^\top x) s a_{s-1}(w^\top x') \frac{u w_{d-1}}{\sqrt{1-u^2}}]$, where $w_{d-1}$ is the normally distributed random variable.

Taking into account the constant normalization of the RF kernel $c^2 = \frac{2}{(2s-1)!!}$, we have

$$
\begin{aligned}
\kappa_s'(u) &= \frac{2}{(2s-1)!!} \frac{\partial}{\partial u} \mathbb{E}_{w \sim \mathcal{N}(\mathbf{0}_d, \mathbf{I}_d)} \left[ a_s(w^\top x) a_s(w^\top x') \right] \\
&= \frac{2 s^2}{(2s-1)!!} \mathbb{E}_{w \sim \mathcal{N}(\mathbf{0}_d, \mathbf{I}_d)} \left[ a_{s-1}(w^\top x) a_{s-1}(w^\top x') \right] \\
&= \frac{s^2}{2s-1} \kappa_{s-1},
\end{aligned}
$$

which proves the lemma.

**Lemma 2 (Stein's Lemma)** *Suppose $X \sim \mathcal{N}(0,1)$ is a normally distributed random variable. Consider a function $g : \mathbb{R} \to \mathbb{R}$ such that both $\mathbb{E}[Xg(X)]$ and $\mathbb{E}[g'(X)]$ exist. We then have*

$$
\mathbb{E}[Xg(X)] = \mathbb{E}[g'(X)].
$$

The proof of Stein's lemma follows from an integration by parts (see, e.g., Stein, 1986; Nourdin & Peccati, 2009; Chen et al., 2011).

# E  PROOF OF PROPOSITION 1

This proposition follows from Theorem 1 of Bietti & Bach (2020), which proved that the eigendecay of a rotationally invariant kernel $\kappa : [-1, 1] \to \infty$ can be determined based on its asymptotic expansions around the endpoints $\pm 1$. Recall that a rotationally invariant kernel $k$ can be represented as $\kappa(x^\top x') = k(x, x')$. Their result is formally given in the following lemma.

**Lemma 3 (Theorem 1 in Bietti & Bach (2020))** *Assume $\kappa : [-1, 1] \to \mathbb{R}$ is $C^\infty$ and has the following asymptotic expansions around $\pm 1$*

$$
\begin{aligned}
\kappa(1 - t) &= p_{+1}(t) + c_{+1}t^\theta + o(t^\theta), \\
\kappa(-1 + t) &= p_{-1}(t) + c_{-1}t^\theta + o(t^\theta),
\end{aligned}
$$

*for $t > 0$, where $p_{\pm 1}$ are polynomials, and $\theta > 0$ is not an integer. Also, assume that the derivatives of $\kappa$ admit similar expansions obtained by differentiating the above ones. Then, there exists $C_{d,\theta}$ such that, when $i$ is even, if $c_{+1} \neq -c_{-1}$: $\tilde{\lambda}_i \sim (c_{+1} + c_{-1})C_{d,\theta}i^{-d-2\theta+1}$, and, when $i$ is odd, if $c_{+1} \neq c_{-1}$: $\tilde{\lambda}_i \sim (c_{+1} - c_{-1})C_{d,\theta}l^{-d-2\theta+1}$. In the case $|c_{+1}| = |c_{-1}|$, then we have $\tilde{\lambda}_i = o(l^{-d-2\theta+1})$ for one of the two parities. If $\kappa$ is infinitely differentiable on $[-1, 1]$ so that no such $\theta$ exists, the $\tilde{\lambda}_i$ decays faster than any polynomial.*

Building on this result, the eigendecays can be obtained by deriving the endpoint expansions for the RF and NT kernels. That is derived in Bietti & Bach (2020) for the ReLU activation function. For completeness, here, we derive the endpoint expansions for RF and NT kernels associated with $s - 1$ times differentiable activation functions $a_s(.)$.

Recall that for a 2 layer network, the corresponding RF and NT kernels are given by

$$
\begin{aligned}
\kappa_{\text{NT},s}(x^\top x') &= c^2(x^\top x')\mathbb{E}_{w \sim \mathcal{N}(\mathbf{0}_d, \mathbf{I}_d)}[a_s'(w^\top x)a_s'(w^\top x')] + \kappa_s(x^\top x'), \\
\kappa_s(x^\top x') &= c^2 \mathbb{E}_{w \sim \mathcal{N}(\mathbf{0}_d, \mathbf{I}_d)}[a_s(w^\top x)a_s(w^\top x')].
\end{aligned}
$$

For the special cases of $s = 0$ and $s = 1$, the following closed form expressions can be derived by taking the expectations.

$$
\begin{aligned}
\kappa_0(u) &:= \mathbb{E}_{w \sim \mathcal{N}(\mathbf{0}_d, \mathbf{I}_d)}[a_0(w^\top x)a_0(w^\top x')] = \frac{1}{\pi}(\pi - \arccos(u)), \\
\kappa_1(u) &= \frac{1}{\pi}\left(u(\pi - \arccos(u)) + \sqrt{1 - u^2}\right),
\end{aligned}
$$

where $u = x^\top x'$. At the endpoints $\pm 1$, $\kappa_0(u), \kappa_1(u)$ have the following asymptotic expansions.

$$
\begin{aligned}
\kappa_0(1 - t) &= 1 - \frac{\sqrt{2}}{\pi}t^{\frac{1}{2}} + o(t^{\frac{1}{2}}), \\
\kappa_0(-1 + t) &= \frac{\sqrt{2}}{\pi}t^{\frac{1}{2}} + o(t^{\frac{1}{2}}), \\
\kappa_1(1 - t) &= 1 - t + \frac{2\sqrt{2}}{3\pi}t^{\frac{3}{2}} + o(t^{\frac{3}{2}}), \\
\kappa_1(-1 + t) &= \frac{2\sqrt{2}}{3\pi}t^{\frac{3}{2}} + o(t^{\frac{3}{2}}).
\end{aligned}
\tag{9}
$$

These endpoint expansions where used in Bietti & Bach (2020) to obtain the eigendecay of the RF and NT kernels with ReLU activation functions.

We first extend the results on the endpoint expansions and the eigendecay to a 2 layer neural network with $s > 1$, in Subsection E.1. Then, we extend the derivation to $l > 2$ layer neural networks, in Subsection E.2. In Subsection E.3, we show how the eigendecays in terms of $\tilde{\lambda}_i$ can be translated to the ones in terms of $\lambda_i$.

### E.1 ENDPOINT EXPANSIONS FOR 2 LAYER NETWORKS WITH $s > 1$

We first note that the normalization constant $c^2 = \frac{2}{(2s-1)!!}$, suggested in Section 2, ensures $\kappa_s(1) = 1$, for all $s \geq 1$. The reason is that, by the well known values for the even moments of the normal distribution, we have

$$\mathbb{E}_{w \sim \mathcal{N}(\mathbf{0}_d, \mathbf{I}_d)}[a_s(w^\top x)a_s(w^\top x')] = \frac{(2s-1)!!}{2},$$

when $x^\top x' = 1$. Also notice that $\kappa_s(-1) = 0$, for all $s \geq 1$. The reason is that when $x^\top x' = -1$, at least one of $w^\top x$ and $w^\top x'$ is non-positive.

We note that the normalization constant is considered only for the convenience of some calculations, and does not affect the exponent in the eigendecays. Specifically, scaling the kernel with a constant factor, scales the corresponding Mercer eigenvalues with the same constant. The constant factor scaling does not affect the Mercer eigenfunctions.

From Lemma 1, recall

$$\kappa_s'(.) = \frac{s^2}{2s-1}\kappa_{s-1}(.).$$

This is a key component in deriving the endpoint expansions for $s > 1$. In particular, this allows us to recursively obtain the endpoint expansions of $\kappa_s(.)$ from those of $\kappa_{s-1}(.)$, by integration. That results in the following expansions for $\kappa_s$.

$$
\begin{aligned}
\kappa_s(1-t) &= p_{+1,s}(t) + c_{+1,s}t^{\frac{2s+1}{2}} + o(t^{\frac{2s+1}{2}}), \\
\kappa_s(-1+t) &= p_{-1,s}(t) + c_{-1,s}t^{\frac{2s+1}{2}} + o(t^{\frac{2s+1}{2}}),
\end{aligned}
\tag{10}
$$

Here, $p_{+1,s}(t) = -\frac{s^2}{2s-1}\int p_{+1,s-1}(t)dt$, subject to $p_{+1,s}(0) = 1$ (that follows form $\kappa_s(1) = 1$). Thus, $p_{+1,s}(t)$ can be obtained recursively, starting from $p_{+1,1}(t) = 1 - t$ (see equation 9). For example,

$$
\begin{aligned}
p_{+1,2}(t) &= -\frac{4}{3}(-\frac{3}{4} + t - \frac{t^2}{2}), \\
p_{+1,3}(t) &= \frac{36}{15}(\frac{15}{36} - \frac{3}{4}t + \frac{1}{2}t^2 - \frac{1}{6}t^3),
\end{aligned}
$$

and so on.

Similarly, $p_{-1,s}(t)$ can be expressed in closed form using $p_{-1,s}(t) = \frac{s^2}{2s-1}\int p_{-1,s-1}(t)dt$ subject to $p_{-1,s}(0) = 0$ (that follows from $\kappa_s(-1) = 0$), and starting from $p_{-1,1}(t) = 0, \forall t$ (see equation 9). That leads to $p_{-1,s}(t) = 0, \forall s > 1, t$.

The exact expressions of $p_{\pm1,s}(t)$ however do not affect the eigendecay. Instead, the constants $c_{\pm1,s}$ are important for the eigendecay, based on Lemma 3. The constants can also be obtained recursively. Starting from $c_{+1,1} = \frac{2\sqrt{2}}{3\pi}$ (see equation 9), $c_{+1,s} = \frac{-2s^2 c_{+1,s-1}}{(2s-1)(2s+1)}$. Also starting from $c_{-1,1} = \frac{2\sqrt{2}}{3\pi}$ (see equation 9), $c_{-1,s} = \frac{2s^2 c_{+1,s-1}}{(2s-1)(2s+1)}$. This recursive relation leads to

$$c_{-1,s} = \frac{2^s\sqrt{2}}{\pi}\prod_{r=1}^{s}\frac{r^2}{(4r^2-1)}, \tag{11}$$

and $c_{+1,s} = (-1)^{s-1}c_{-1,s}$.

In the case of RF kernel, applying Lemma 3, we have $\tilde{\lambda}_i \sim i^{-d-2s}$, for $i$ having the opposite parity of $s$, and $\tilde{\lambda}_i = o(i^{-d-2s})$ for $i$ having the same parity of $s$.

**For the NT kernel**, using 2, we have

$$
\begin{aligned}
\kappa_{\text{NT},s}(1-t) &= c^2(1-t)s^2\kappa_{s-1}(1-t) + \kappa_s(1-t) \\
\kappa_{\text{NT},s}(-1+t) &= c^2(-1+t)s^2\kappa_{s-1}(-1+t) + \kappa_s(-1+t)
\end{aligned}
\tag{12}
$$

Using the expansion of the RF kernel, we get

$$
\begin{aligned}
\kappa_{\mathrm{NT},s}(1-t) &= p''_{+1,s}(t) + c''_{+1,s}t^{\frac{2s-1}{2}} + o(t^{\frac{2s-1}{2}}), \\
\kappa_{\mathrm{NT},s}(-1+t) &= p''_{-1,s}(t) + c''_{-1,s}t^{\frac{2s-1}{2}} + o(t^{\frac{2s-1}{2}}),
\end{aligned} \tag{13}
$$

where

$$
\begin{aligned}
p''_{+1,s}(t) &= c^2(1-t)s^2 p_{+1,s-1}(t) + p_s(t), \\
p''_{-1,s}(t) &= c^2(-1+t)s^2 p_{-1,s}(t) + p_{-1,s}(-1+t),
\end{aligned}
$$

and

$$
\begin{aligned}
c''_{+1,s} &= c_{+1,s-1}, \\
c''_{-1,s} &= c_{-1,s-1}.
\end{aligned}
$$

Thus, in the case of NT kernel, applying Lemma 3, we have $\tilde{\lambda}_i \sim i^{-d-2s+2}$, for $i$ having the same parity of $s$, and $\tilde{\lambda}_i = o(i^{-d-2s+2})$ for $i$ having the opposite parity of $s$.

### E.2 ENDPOINT EXPANSIONS FOR $l > 2$ LAYER NETWORKS

We here extend the endpoint expansions and the eigendecays to $l > 2$ later networks. Recall $\kappa_s^l(u) = \kappa_s(\kappa_s^{l-1}(u))$. Using this recursive relation over $l$, we prove the following expressions

$$
\begin{aligned}
\kappa_s^l(1-t) &= p^l_{+1,s}(t) + c^l_{+1,s}t^{\frac{2s+1}{2}} + o(t^{\frac{2s+1}{2}}) \\
\kappa_s^l(-1+t) &= p^l_{-1,s}(t) + c^l_{-1,s}t^{\frac{2s+1}{2}} + o(t^{\frac{2s+1}{2}})
\end{aligned}
$$

where $p^l_{\pm 1,s}$ are polynomials and $c^l_{\pm 1,s}$ are constants.

The notations $p_{+1,s}$ and $c_{+1,s}$ in Subsection E.1 correspond to $p^2_{+1,s}$ $c^2_{+1,s}$, where we drop the superscript specifying the number $l$ of layers, for 2 layer networks.

For the endpoint expansion at 1, we have

$$
\begin{aligned}
\kappa_s^l(1-t) &= \kappa_s(\kappa_s^{l-1}(1-t)) \\
&= \kappa_s(1 - 1 + p^{l-1}_{+1,s}(t) + c^{l-1}_{+1,s}t^{\frac{2s+1}{2}} + o(t^{\frac{2s+1}{2}})) \\
&= p_{+1,s}(1 - p^{l-1}_{+1,s}(t) - c^{l-1}_{+1,s}t^{\frac{2s+1}{2}} - o(t^{\frac{2s+1}{2}})) \\
&\quad + c_{+1,s}(1 - p^{l-1}_{+1,s}(t) - c^{l-1}_{+1,s}t^{\frac{2s+1}{2}} - o(t^{\frac{2s+1}{2}}))^{\frac{2s+1}{2}} \\
&\quad + o\left((1 - p^{l-1}_{+1,s}(t) - c^{l-1}_{+1,s}t^{\frac{2s+1}{2}} - o(t^{\frac{2s+1}{2}}))^{\frac{2s+1}{2}}\right)
\end{aligned}
$$

Thus, we have

$$
c^l_{+1,s} = (-q^{l-1,1}_{+1,s})^{\frac{2s+1}{2}} c_{+1,s} - q^{2,1}_{+1,s}c^{l-1}_{+1,s}, \tag{14}
$$

where $q^{l,i}_{+1,s}$ is the coefficient of $t^i$ in $p^l_{+1,s}(t)$. For these coefficients, we have

$$
q^{l,1}_{+1,s} = -q^{2,1}_{+1,s}q^{l-1,1}_{+1,s}. \tag{15}
$$

Starting from $q^{2,1}_{+1,s} = -\frac{s^2}{2s-1}$ (which can be seen from the recursive expression of $p_{+1,s}$ given in Section E.1), we get $q^{l,1}_{+1,s} = -\frac{s^{2(l-1)}}{(2s-1)^{l-1}}$. We thus have

$$
c^l_{+1,s} = \left(\frac{s^{(2s+1)}}{(2s-1)^{\frac{(2s+1)}{2}}}\right)^{l-2} c_{+1,s} + \frac{s^2}{2s-1}c^{l-1}_{+1,s}. \tag{16}
$$

That implies

$$
c^l_{+1,s} \sim \left(\frac{s^{(2s+1)}}{(2s-1)^{\frac{(2s+1)}{2}}}\right)^{l-2} c_{+1,s}, \tag{17}
$$

when $s > 1$.

The characterization of $c_{+1,s}^l$ shows that our results do not apply to deep neural networks when $s > 1$, in the sense that the constants grow exponentially in $l$ (as stated in Remark 2).

For the endpoint expansion at $-1$, we have

$$
\begin{aligned}
\kappa_s^l(-1+t) &= \kappa_s(\kappa_s^{l-1}(-1+t)) \\
&= \kappa_s(p_{-1,s}^{l-1}(t) + c_{-1,s}^{l-1} t^{\frac{2s+1}{2}} + o(t^{\frac{2s+1}{2}})) \\
&= \kappa_s(q_{-1,s}^{l-1,0}) + \kappa_s'(q_{-1,s}^{l-1,0})(p_{-1,s}^{l-1}(t) - q_{-1,s}^{l-1,0} + c_{-1,s}^{l-1} t^{\frac{2s+1}{2}}) + o(t^{\frac{2s+1}{2}}).
\end{aligned}
$$

where $q_{-1,s}^{l,i}$ is the coefficient of $t^i$ in $p_{-1,s}^l(t)$. From the expression above we can see that

$$
q_{-1,s}^{l,0} = \kappa_s(q_{-1,s}^{l-1,0}) \tag{18}
$$

Thus, $0 \le q_{-1,s}^{l,0} \le 1$. In addition, the expression above implies

$$
c_{-1,s}^l = c_{-1,s}^{l-1} \kappa_s'(q_{-1,s}^{l-1,0}). \tag{19}
$$

From Lemma 1, $\kappa_s'(u) \le \frac{s^2}{2s-1}$, for all $u$. Therefore,

$$
\begin{aligned}
c_{-1,s}^l &\le \left(\frac{s^2}{2s-1}\right)^{l-2} c_{-1,s} \\
&= o(c_{+1,s}^l),
\end{aligned}
$$

when, $s > 1$.

Comparing $c_{\pm 1,a}^l$, we can see that for $l > 2$, we have $|c_{+1,s}^l| \ne |c_{-1,s}^l|$. Thus, for the RF kernel with $l > 2$, applying Lemma 3, we have $\tilde{\lambda}_i \sim i^{-d-2s}$.

**For the NT kernel,** recall

$$
\kappa_{\mathrm{NT},s}^l(u) = c^2 \kappa_{\mathrm{NT},s}^{l-1}(u) \kappa_s'(\kappa_s^{l-1}(u)) + \kappa_s(u).
$$

The second term is exactly the same as the RF kernel. Recall the expression of $\kappa_s'$ based on $\kappa_{s-1}$ from Lemma 1. To find the endpoint expansions of the first term, we prove the following expressions for $\kappa_{s-1}(\kappa_s^{l-1}(u))$.

$$
\begin{aligned}
\kappa_{s-1}(\kappa_s^{l-1}(1-t)) &= p_{+1,s}^{\prime l}(t) + c_{+1,s}^{\prime l} t^{\frac{2s-1}{2}} + o(t^{\frac{2s-1}{2}}) \\
\kappa_{s-1}(\kappa_s^{l-1}(-1+t)) &= p_{-1,s}^{\prime l}(t) + c_{-1,s}^{\prime l} t^{\frac{2s+1}{2}} + o(t^{\frac{2s+1}{2}})
\end{aligned}
$$

We use the notation used for the RF kernel to write the following expansion around 1

$$
\begin{aligned}
\kappa_{s-1}(\kappa_s^{l-1}(1-t)) &= \kappa_{s-1}(1 - 1 + p_{+1,s}^{l-1}(t) + c_{+1,s}^{l-1} t^{\frac{2s+1}{2}} + o(t^{\frac{2s+1}{2}})) \\
&= p_{+1,s-1}(1 - p_{+1,s}^{l-1}(t) - c_{+1,s}^{l-1} t^{\frac{2s+1}{2}} - o(t^{\frac{2s+1}{2}})) \\
&\quad + c_{+1,s-1}(1 - p_{+1,s}^{l-1}(t) - c_{+1,s}^{l-1} t^{\frac{2s+1}{2}} - o(t^{\frac{2s+1}{2}}))^{\frac{2s-1}{2}} \\
&\quad + o\left((1 - p_{+1,s}^{l-1}(t) - c_{+1,s}^{l-1} t^{\frac{2s+1}{2}} - o(t^{\frac{2s+1}{2}}))^{\frac{2s-1}{2}}\right).
\end{aligned}
$$

Thus, we have

$$
c_{+1,s}^{\prime l} = (-q_{+1,s}^{l-1,1})^{\frac{2s-1}{2}} c_{+1,s-1} - q_{+1,s-1}^{2,1} c_{+1,s}^{l-1} \tag{20}
$$

Recall, form the analysis of the RF kernel that $q_{+1,s}^{l,1} = -\frac{s^2(l-1)}{(2s-1)^{l-1}}$. We thus have

$$
c_{+1,s}^{\prime l} = \left(\frac{s^{(2s-1)}}{(2s-1)^{\frac{(2s-1)}{2}}}\right)^{l-2} c_{+1,s-1} + \frac{(s-1)^2}{2s-3} c_{+1,s}^{l-1}. \tag{21}
$$

That implies

$$c_{+1,s}^{\prime l} \sim \left( \frac{s^{(2s-1)}}{(2s-1)^{\frac{(2s-1)}{2}}} \right)^{l-2} c_{+1,s-1}. \tag{22}$$

For the endpoint expansion at $-1$, we have

$$
\begin{aligned}
\kappa_{s-1}(\kappa_s^{l-1}(-1+t)) &= \kappa_{s-1}(p_{-1,s}^{l-1}(t) + c_{-1,s}^{l-1} t^{\frac{2s+1}{2}} + o(t^{\frac{2s+1}{2}})) \\
&= \kappa_{s-1}(q_{-1,s}^{l-1,0}) + \kappa_{s-1}'(q_{-1,s}^{l-1,0})(p_{-1,s}^{l-1}(t) - q_{-1,s}^{l-1,0} + c_{-1,s}^{l-1} t^{\frac{2s+1}{2}}) + o(t^{\frac{2s+1}{2}}).
\end{aligned}
$$

We thus have

$$c_{-1,s}^{\prime l} = c_{-1,s-1}^{l-1} \kappa_{s-1}'(q_{-1,s}^{l-1,0}). \tag{23}$$

Since $\kappa_{s-1}' \le \frac{(s-1)^2}{2s-3}$, we have

$$c_{-1,s}^{\prime l} \le \left( \frac{s^2}{2s-3} \right)^{l-2} c_{-1,s-1}.$$

Now we use the end point expansions of $\kappa_s^l$ and $\kappa_{s-1}(\kappa_s^{l-1})$ to obtain the following endpoint expansions for $\kappa_{\mathrm{NT},s}^l$.

$$
\begin{aligned}
\kappa_{\mathrm{NT},s}^l(1-t) &= p_{+1,s}^{\prime\prime l}(t) + c_{+1,s}^{\prime\prime l} t^{\frac{2s-1}{2}} + o(t^{\frac{2s-1}{2}}), \\
\kappa_{\mathrm{NT},s}^l(-1+t) &= p_{-1,s}^{\prime\prime l}(t) + c_{-1,s}^{\prime\prime l} t^{\frac{2s-1}{2}} + o(t^{\frac{2s-1}{2}}).
\end{aligned}
$$

We have

$$c_{+1,s}^{\prime\prime l} = \frac{c^2 s^2}{2s-1} q_{+1,s}^{\prime l,0} c_{+1,s}^{\prime\prime l-1} + \frac{c^2 s^2}{2s-1} q_{+1,s}^{\prime\prime l-1,0} c_{+1,s}^{\prime l},$$

and,

$$q_{+1,s}^{\prime\prime l,0} = \frac{c^2 s^2}{2s-1} q_{+1,s}^{\prime\prime l-1,0} q_{+1,s}^{\prime l,0} + q_{+1,s}^{l,0}. \tag{24}$$

Since $\kappa_s(1) = 1$, by induction we have $\kappa_s^l(1) = 1$, $\forall l \ge 2$. In addition, $\kappa_{s-1}(\kappa_s^{l-1}(1)) = \kappa_{s-1}(1) = 1$, $\forall l \ge 3$. Therefore $q_{+1,s}^{\prime l,0}, q_{+1,s}^{l,0} = 1$. Thus,

$$q_{+1,s}^{\prime\prime l,0} = \frac{c^2 s^2}{2s-1} q_{+1,s}^{\prime\prime l-1,0} + 1. \tag{25}$$

Starting from $q_{+1,s}^{\prime\prime 2,0} = \frac{s^2}{2s-1} + 1$, we can derive the following expression for $q_{+1,s}^{\prime\prime l,0}$, when $s > 1$,

$$
\begin{aligned}
q_{+1,s}^{\prime\prime l,0} &= \frac{1}{c^2} \left( \frac{c^2 s^2}{2s-1} \right)^{l-1} + \frac{\left( \frac{c^2 s^2}{2s-1} \right)^{l-1} - 1}{\left( \frac{c^2 s^2}{2s-1} \right) - 1} \\
&\sim \frac{1}{c^2} \left( \frac{c^2 s^2}{2s-1} \right)^{l-1}.
\end{aligned}
$$

When $s = 1$, $q_{+1,s}^{\prime\prime l,0} = l$, that is consistent with the results in Bietti & Bach (2020).

For $c_{+1,s}^{\prime\prime l}$, we thus have

$$c_{+1,s}^{\prime\prime l} = \frac{c^2 s^2}{2s-1} c_{+1,s}^{\prime\prime l-1} + \frac{c^2 s^2}{2s-1} q_{+1,s}^{\prime\prime l-1,0} c_{+1,s}^{\prime l}.$$

Replacing the expressions for $q_{+1,s}^{\prime\prime\prime l-1,0}$ and $c_{+1,s}^{\prime l}$ derived above, we obtain

$$c_{+1,s}^{\prime\prime l} \sim (\frac{1}{c^2})^{(\frac{2s-1}{2})(l-2)+1} \left(\frac{c^2 s^2}{2s-1}\right)^{(\frac{2s+1}{2})(l-2)+1} c_{+1,s-1}. \tag{26}$$

For the constant in the expansion around $-1$, we have

$$c_{-1,s}^{\prime\prime l} = \frac{c^2 s^2}{2s-1} q_{-1,s}^{\prime l,0} c_{-1,s}^{\prime\prime l-1}, \tag{27}$$

where $q_{-1,s}^{\prime l,0} = \kappa_{s-1}(q_{-1,s}^{l-1,0})$ is bounded between 0 and 1. Thus, starting from $c_{-1,s}^{\prime\prime 2} = c_{-1,s-1}$, we obtain

$$c_{-1,s}^{\prime\prime l} \leq \left(\frac{c^2 s^2}{2s-1}\right)^{l-2} c_{-1,s-1} \tag{28}$$

Comparing $c_{\pm 1,a}^{\prime\prime l}$, we can see that for $l > 2$, we have $|c_{+1,s}^{\prime\prime l}| \neq |c_{-1,s}^{\prime\prime l}|$. Thus, for the NT kernel with $l > 2$, applying Lemma 3, we have $\tilde{\lambda}_i \sim i^{-d-2s+2}$.

### E.3   THE EIGENDECAYS IN TERMS OF $\lambda_i$

Recall the multiplicity $N_{d,i} = \frac{2i+d-2}{i}\binom{i+d-3}{d-2}$ of $\tilde{\lambda}_i$. Here we take into account this multiplicity to give the eigendecay expressions in terms of $\lambda_i$.

Using $(\frac{n}{k})^k \leq \binom{n}{k} \leq (\frac{ne}{k})^k$, for all $k, n \in \mathbb{N}$, we have, for all $i > 1$ and $d > 2$,

$$2(i-1)^{d-2}(\frac{1}{d-2} + \frac{1}{i-1})^{d-2} \leq N_{d,i} \leq \frac{(d+2)e^{d-2}}{2}(i-1)^{d-2}(\frac{1}{d-2} + \frac{1}{i-1})^{d-2},$$

where we used $2 < \frac{2i+d-2}{i} < \frac{d+2}{2}$. We thus have $N_{d,i} \sim (i-1)^{d-2}$, for the scaling of $N_{d,i}$ with $i$, with constants given above.

Recall we define $\lambda_i = \tilde{\lambda}_{i'}$, for $i$ and $i'$ which satisfy $\sum_{i''=1}^{i'-1} N_{d,i''} < i \leq \sum_{i''=1}^{i'} N_{d,i''}$. Using $N_{d,i} \sim (i-1)^{d-2}$, we have $\sum_{i''=1}^{i'} N_{d,i''} \sim i'^{d-1}$. Thus $\lambda_i \sim \tilde{\lambda}_{i'} = \tilde{\lambda}_{i^{\frac{1}{d-1}}}$. Replacing $i$ with $i^{\frac{1}{d-1}}$, in the expressions derived for $\tilde{\lambda}_i$ in this section, we obtain the eigendecay expressions for $\lambda_i$ reported in Table 2.

## F   PROOF OF THEOREM 1

The Matérn kernel can also be decomposed in the basis of spherical harmonics. In particular, Borovitskiy et al. (2020) proved the following expression for the Matérn kernel with smoothness parameter $\nu$ on the hypersphere $\mathbb{S}^{d-1}$ (see, also Dutordoir et al., 2020, Appendix B)

$$k_\nu(x, x') = \sum_{i=1}^{\infty} \sum_{j=1}^{N_{d,i}} (\frac{2\nu}{\kappa^2} + i(i+d-2))^{-(\nu + \frac{d-1}{2})} \tilde{\phi}_{i,j}(x)\tilde{\phi}_{i,j}(x'),$$

where $\tilde{\phi}_{i,j}$ are the spherical harmonics.

Theorem 5 implies that the RKHS of Metérn kernel is also constructed as a span of spherical harmonics. Thus, in order to show the equivalence of the RKHSs of the neural kernels with various activation functions and a Matérn kernel with the corresponding smoothness, we show that the ratio between their norms is bounded by absolute constants.

Let $f \in \mathcal{H}^l_{k_{\mathrm{NT},s}}$, with $l \geq 2$. As a result of Mercer's representation theorem, we have

$$
\begin{aligned}
f(\cdot) &= \sum_{i=1}^{\infty} \sum_{j=1}^{N_{d,i}} w_{i,j} \tilde{\lambda}_i^{\frac{1}{2}} \tilde{\phi}_{i,j}(\cdot) \\
&= \sum_{i=1}^{\infty} \sum_{j=1}^{N_{d,i}} w_{i,j} \frac{\tilde{\lambda}_i^{\frac{1}{2}}}{(\frac{2\nu}{\kappa^2} + i(i+d-2))^{-\frac{1}{2}(\nu+\frac{d-1}{2})}} \left(\frac{2\nu}{\kappa^2} + i(i+d-2)\right)^{-\frac{1}{2}(\nu+\frac{d-1}{2})} \tilde{\phi}_{i,j}(\cdot).
\end{aligned}
$$

Note that

$$
\frac{\tilde{\lambda}_i}{(\frac{2\nu}{\kappa^2} + i(i+d-2))^{-(\nu+\frac{d-1}{2})}} = \mathcal{O}\left(\frac{\tilde{\lambda}_i}{i^{-2\nu-d+1}}\right).
$$

For the NT kernel, from Proposition 1, $\tilde{\lambda}_i = \mathcal{O}(i^{-d-2s+2})$. Thus, when $\nu = s - \frac{1}{2}$,

$$
\frac{\tilde{\lambda}_i}{(\frac{2\nu}{\kappa^2} + i(i+d-2))^{-(\nu+\frac{d-1}{2})}} = \mathcal{O}(1).
$$

So, with $\nu = s - \frac{1}{2}$ we have

$$
\begin{aligned}
\|f\|^2_{\mathcal{H}_{k_\nu}} &= \sum_{i=1}^{\infty} \sum_{j=1}^{N_{d,i}} w_{i,j}^2 \frac{\tilde{\lambda}_i}{(\frac{2\nu}{\kappa^2} + i(i+d-2))^{-(\nu+\frac{d-1}{2})}} \\
&= \mathcal{O}\left(\sum_{i=1}^{\infty} \sum_{j=1}^{N_{d,i}} w_{i,j}^2\right) \\
&= \mathcal{O}\left(\|f\|^2_{\mathcal{H}_{k_{\mathrm{NT},s}^l}}\right).
\end{aligned}
$$

That proves $\mathcal{H}_{k_{\mathrm{NT},s}^l} \subset \mathcal{H}_{k_\nu}$.

A similar proof shows that when $\nu = s + \frac{1}{2}$, $\mathcal{H}^l_{k_s} \subset \mathcal{H}_{k_\nu}$.

Now, let $f \in \mathcal{H}_{k_\nu}$. As a result of Mercer's representation theorem, we have

$$
\begin{aligned}
f(\cdot) &= \sum_{i=1}^{\infty} \sum_{j=1}^{N_{d,i}} w_{i,j} \left(\frac{2\nu}{\kappa^2} + i(i+d-2)\right)^{-\frac{1}{2}(\nu+\frac{d-1}{2})} \tilde{\phi}_{i,j}(\cdot) \\
&= \sum_{i=1}^{\infty} \sum_{j=1}^{N_{d,i}} w_{i,j} \frac{(\frac{2\nu}{\kappa^2} + i(i+d-2))^{-\frac{1}{2}(\nu+\frac{d-1}{2})}}{\tilde{\lambda}_i^{\frac{1}{2}}} \tilde{\lambda}_i^{\frac{1}{2}} \tilde{\phi}_{i,j}(\cdot).
\end{aligned}
$$

For the NT kernel with $l > 2$, from Proposition 1, $\tilde{\lambda}_l \sim l^{-d-2s+2}$. Thus, when $\nu = s - \frac{1}{2}$,

$$
\frac{(\frac{2\nu}{\kappa^2} + l(l+d-2))^{-(\nu+\frac{d-1}{2})}}{\tilde{\lambda}_l} = \mathcal{O}(1).
$$

So, with $\nu = s - \frac{1}{2}$ we have

$$
\begin{aligned}
\|f\|^2_{\mathcal{H}_{k_{\mathrm{NT},s}^l}} &= \sum_{i=1}^{\infty} \sum_{j=1}^{N_{d,i}} w_{i,j}^2 \frac{(\frac{2\nu}{\kappa^2} + i(i+d-2))^{-(\nu+\frac{d-1}{2})}}{\tilde{\lambda}_i} \\
&= \mathcal{O}\left(\sum_{i=1}^{\infty} \sum_{j=1}^{N_{d,i}} w_{i,j}^2\right) \\
&= \mathcal{O}\left(\|f\|^2_{\mathcal{H}_{k_\nu}}\right).
\end{aligned}
$$

That proves, for $l > 2$, when $\nu = s - \frac{1}{2}$, $\mathcal{H}_{k_\nu} \subset \mathcal{H}^l_{k_{\mathrm{NT},s}}$. A similar proof shows that for $l > 2$, when $\nu = s + \frac{1}{2}$, $\mathcal{H}_{k_\nu} \subset \mathcal{H}^l_{k_s}$, which completes the proof of Theorem 1.

## G  PROOF OF THEOREM 2

Recall the definition of the information gain for a kernel $\kappa$,

$$\mathcal{I}(Y_n; F) = \frac{1}{2} \log \det \left( \boldsymbol{I}_n + \frac{1}{\lambda^2} \mathbf{K}_n \right).$$

To bound the information gain, we define two new kernels resulting from the partitioning of the eigenvalues of the neural kernels to the first $M$ eigenvalues and the remainder. In particular, we define

$$
\begin{aligned}
\tilde{\kappa}(x^\top x) &= \sum_{i=1}^{M} \sum_{j=1}^{N_{d,i}} \tilde{\lambda}_i \tilde{\phi}_{i,j}(x) \tilde{\phi}_{i,j}(x'), \\
\tilde{\tilde{\kappa}}(x^\top x) &= \sum_{i=M+1}^{\infty} \sum_{j=1}^{N_{d,i}} \tilde{\lambda}_i \tilde{\phi}_{i,j}(x) \tilde{\phi}_{i,j}(x').
\end{aligned}
\tag{29}
$$

We present the proof for the NT kernel. A similar proof applies to the RF kernel.

The truncated kennel at $M$ eigenvalues, $\tilde{\kappa}(x^\top x)$, corresponds to the projection of the RKHS of $\kappa$ onto a finite dimensional space with dimension $\sum_{i=1}^{M} \sum_{j=1}^{N_{d,i}}$.

We use the notations $\tilde{\mathbf{K}}_n$ and $\tilde{\tilde{\mathbf{K}}}_n$ to denote the kernel matrices corresponding to $\tilde{\kappa}$ and $\tilde{\tilde{\kappa}}$, respectively.

As proposed in Vakili et al. (2021b), we write

$$
\begin{aligned}
\log \det \left( \mathbf{I}_n + \frac{1}{\lambda^2} \mathbf{K}_n \right) &= \log \det \left( \mathbf{I}_n + \frac{1}{\lambda^2} (\tilde{\mathbf{K}}_n + \tilde{\tilde{\mathbf{K}}}_n) \right) \\
&= \log \det \left( \mathbf{I}_n + \frac{1}{\lambda^2} \tilde{\mathbf{K}}_n \right) + \log \det \left( \mathbf{I}_n + \frac{1}{\lambda^2} (\mathbf{I}_n + \frac{1}{\lambda^2} \tilde{\mathbf{K}}_n)^{-1} \tilde{\tilde{\mathbf{K}}}_n \right).
\end{aligned}
\tag{30}
$$

The analysis in Vakili et al. (2021b) crucially relies on an assumption that the eigenfunctions are uniformly bounded. In the case of spherical harmonics however (see, e.g., Stein & Weiss, 2016, Corollary 2.9)

$$
\sup_{\substack{i=1,2,\ldots, \\ j=1,2,\ldots,N_{d,i}, \\ x \in \mathcal{X}}} \tilde{\phi}_{i,j}(x) = \infty.
\tag{31}
$$

Thus, their analysis does not apply to the case of neural kernels on the hypersphere.

To bound the two terms on the right hand side of equation 30, we first prove in Lemma 4 that $\tilde{\tilde{\kappa}}(x, x')$ approaches $0$ as $M$ grows (uniformly in $x$ and $x'$), with a certain rate.

**Lemma 4** *For the NT kernel, we have $\tilde{\tilde{\kappa}}(x^\top x') = \mathcal{O}(M^{-2s+1})$, for all $x, x' \in \mathcal{X}$.*

*Proof of Lemma 4:* Recall that the eigenspace corresponding to $\tilde{\lambda}_i$ has dimension $N_{d,i}$ and consists of spherical harmonics of degree $i$. Legendre addition theorem (Maleček & Nádeník, 2001) states

$$
\sum_{j=1}^{N_{d,i}} \phi_{i,j}(x) \phi_{i,j}(x') = c_{i,d} C_i^{(d-2)/2}(\cos(\mathrm{d}(x, x'))),
\tag{32}
$$

where $\mathtt{d}$ is the geodesic distance on the hypersphere, $C_i^{(d-2)/2}$ are Gegenbauer polynomials (those are the same as Legendre polynomials up to a scaling factor), and the constant $c_{i,d}$ is

$$c_{i,d} = \frac{N_{d,i}\Gamma((d-2)/2)}{2\pi^{(d-2)/2}C_k^{(d-2)/2}(1)}. \tag{33}$$

We thus have

$$
\begin{aligned}
\tilde{\tilde{\kappa}}(x^\top x') &= \sum_{i=M+1}^{\infty} \tilde{\lambda}_i \sum_{j=1}^{N_{d,i}} \phi_{i,j}(x)\phi_{i,j}(x') \\
&= \sum_{i=M+1}^{\infty} \tilde{\lambda}_i c_{i,d} C_i^{(d-2)/2}(\cos(\mathtt{d}(x,x'))) \\
&\leq \sum_{i=M+1}^{\infty} \tilde{\lambda}_i N_{d,i} \frac{\Gamma((d-2)/2)}{2\pi^{((d-2)/2)}} \\
&\sim \sum_{i=M+1}^{\infty} i^{-d-2s+2} i^{d-2} \\
&\sim M^{-2s+1}.
\end{aligned}
$$

Here, the inequality follows from $C_i^{(d-2)/2}(\cos(\mathtt{d}(x,x'))) \leq C_i^{(d-2)/2}(1)$, because the Gegenbauer polynomials attain their maximum at the endpoint 1. For the fourth line we used $N_{d,i} \sim i^{d-2}$. The implied constants include the implied constants in $N_{d,i} \sim i^{d-2}$ given in Section E.3, and $\frac{\Gamma((d-2)/2)}{2\pi^{((d-2)/2)}}$. $\square$

We also introduce a notation $N_M = \sum_{i=1}^{M} N_{d,i}$ for the number of eigenvalues corresponding to the spherical harmonics of degree up to $M$, taking into account their multiplicities, that satisfies $N_M \sim M^{d-1}$ (see Section E.3).

We are now ready to bound the two terms on the right hand side of equation 30. Let us define $\boldsymbol{\Phi}_{n,N_M} = [\boldsymbol{\phi}_{N_M}(x_1), \boldsymbol{\phi}_{N_M}(x_2), \dots, \boldsymbol{\phi}_{N_M}(x_n)]^\top$, an $n \times N_M$ matrix which stacks the feature vectors $\boldsymbol{\phi}_{N_M}(x_i) = [\phi_j(x_i)]_{j=1}^{N_M}$, $i = 1, \dots, n$, at the observation points, as its rows. Notice that

$$\tilde{\mathbf{K}}_n = \boldsymbol{\Phi}_{n,N_M} \Lambda_{N_M} \boldsymbol{\Phi}_{n,N_M}^\top,$$

where $\Lambda_{N_M}$ is the diagonal matrix of the eigenvalues defined as $[\Lambda_{N_M}]_{i,j} = \lambda_i \delta_{i,j}$.

Now, consider the Gram matrix

$$\boldsymbol{G} = \Lambda_{N_M}^{\frac{1}{2}} \boldsymbol{\Phi}_{n,N_M}^\top \boldsymbol{\Phi}_{n,N_M} \Lambda_{N_M}^{\frac{1}{2}}.$$

As it was shown in Vakili et al. (2021b), by matrix determinant lemma, we have

$$
\begin{aligned}
\log\det(\mathbf{I}_n + \frac{1}{\lambda^2}\tilde{\mathbf{K}}_n) &= \log\det(\mathbf{I}_{N_M} + \frac{1}{\lambda^2}\boldsymbol{G}) \\
&\leq N_M \log\left(\frac{1}{N_M}\operatorname{tr}(\mathbf{I}_{N_M} + \frac{1}{\lambda^2}\boldsymbol{G})\right) \\
&= N_M \log(1 + \frac{n}{\lambda^2 N_M}).
\end{aligned}
$$

To upper bound the second term on the right hand side of equation 30, we use Lemma 4. In particular, since

$$\operatorname{tr}\left((\mathbf{I}_n + \frac{1}{\lambda^2}\tilde{\mathbf{K}}_n)^{-1}\tilde{\tilde{\mathbf{K}}}_n\right) \leq \operatorname{tr}(\tilde{\tilde{\mathbf{K}}}_n),$$

and $[\tilde{\tilde{\mathbf{K}}}_n]_{i,i} = \mathcal{O}(M^{-2s+1})$, we have

$$\operatorname{tr}\left(\mathbf{I}_n + \frac{1}{\lambda^2}(\mathbf{I}_n + \frac{1}{\lambda^2}\tilde{\mathbf{K}}_n)^{-1}\tilde{\tilde{\mathbf{K}}}_n\right) = \mathcal{O}\left(n(1 + \frac{1}{\lambda^2}M^{-2s+1})\right).$$

Therefore

$$
\begin{aligned}
\log\det\left(\mathbf{I}_n + \frac{1}{\lambda^2}(\mathbf{I}_n + \frac{1}{\lambda^2}\tilde{\mathbf{K}}_n)^{-1}\tilde{\tilde{\mathbf{K}}}_n\right) &\leq n\log\left(\mathcal{O}(1 + \frac{1}{\lambda^2}M^{-2s+1})\right) \\
&\leq \frac{nM^{-2s+1}}{\lambda^2} + \mathcal{O}(1),
\end{aligned}
$$

where for the last line we used $\log(1+z) \leq z$ which holds for all $z \in \mathbb{R}$.

We thus have $\gamma_k(n) = \mathcal{O}(M^{d-1}\log(n) + nM^{-2s+1})$. Choosing $M \sim n^{\frac{1}{d+2s-2}}(\log(n))^{\frac{-1}{d+2s-2}}$, we obtain

$$
\gamma_{\kappa_{\mathrm{NT},s}^l}(n) = \mathcal{O}\left(n^{\frac{d-1}{d+2s-2}}(\log(n))^{\frac{2s-1}{d+2s-2}}\right). \tag{34}
$$

For example, with ReLU activation functions, we have

$$
\gamma_{\kappa_{\mathrm{NT},1}^l}(n) = \mathcal{O}\left(n^{\frac{d-1}{d}}(\log(n))^{\frac{1}{d}}\right).
$$

## H  PROOF OF THEOREM 3

As stated in the paper, the proof follows the same steps as in the proof of Theorem 3 of Vakili et al. (2021a). Their theorem holds for general kernels, provided the bound on MIG. We have adopted their theorem for the special case of neural kernels, inserting our novel bounds on MIG of the neural kernels given in Theorem 2. For completeness, we include a detailed proof here.

The proof consists of tow components. First, we bound the uncertainty estimate $\sigma_n(x)$ in terms of $\gamma_k(n)$. Our novel bounds on $\gamma_k(n)$ of neural kernels given in Theorem 2 allow us to derive explicit bounds on $\sigma_n(x)$. Then, we bound the error $|f(x) - \hat{f}_n(x)|$ in terms of $\sigma_n(x)$, using implicit error bounds.

Recall the way that the dataset $\tilde{\mathcal{D}}_n$ is collected: $x_i = \arg\max_{x \in \mathcal{X}} \sigma_{i-1}(x)$. This ensures that, $\forall x \in \mathcal{X}$, $\sigma_{i-1}(x) \leq \sigma_{i-1}(x_i)$. Due to positive definiteness of the kernel matrix, conditioning on a larger dataset reduces the uncertainty. Thus $\forall x \in \mathcal{X}$ and $\forall i \leq n$, $\sigma_n(x) \leq \sigma_i(x)$. Therefore $\sigma_n^2(x)$ is upper bounded by the average of $\sigma_{i-1}^2(x_i)$ over $i$: $\forall x \in \mathcal{X}$

$$
\sigma_n^2(x) \leq \frac{1}{n}\sum_{i=1}^n \sigma_{i-1}^2(x_i). \tag{35}
$$

It is shown that (e.g., see, Srinivas et al., 2010, Lemmas 5.3, 5.4)

$$
\sum_{i=1}^n \sigma_{i-1}^2(x_i) \leq \frac{2\mathcal{I}(Y_n; F)}{\log(1 + \frac{1}{\lambda^2})}. \tag{36}
$$

Thus, combining equation 35 and equation 36, and by definition of $\gamma_k(n)$, we have, $\forall x \in \mathcal{X}$

$$
\sigma_n(x) \leq \sqrt{\frac{2\gamma_k(n)}{n\log(1 + \frac{1}{\lambda^2})}}. \tag{37}
$$

Now, inserting our bounds on $\gamma_{\kappa_{\mathrm{NT},s}^l}(n)$ and $\gamma_{\kappa_s^l}(n)$ of the neural kernels from Theorem 2, we obtain, $\forall x \in \mathcal{X}$

$$
\begin{aligned}
\sigma_n(x) &\leq n^{\frac{-2s+1}{2d+4s-4}}(\log(n))^{\frac{2s-1}{2d+4s-4}}\sqrt{\frac{2}{\log(1 + \frac{1}{\lambda^2})}}, \quad \text{in the case of NT kernel,} \\
\sigma_n(x) &\leq n^{\frac{-2s-1}{2d+4s}}(\log(n))^{\frac{2s+1}{2d+4s}}\sqrt{\frac{2}{\log(1 + \frac{1}{\lambda^2})}}, \quad \text{in the case of RF kernel.}
\end{aligned} \tag{38}
$$

The second component of the proof is summarized in the following lemma.

**Lemma 5** *Under Assumption 1, we have, with probability at least $1 - \delta$, $\forall x \in \mathcal{X}$*

$$|f(x) - \hat{f}_n(x)| \leq \left( B + C(\log(\frac{n^{d-1}}{\delta}))^{\frac{1}{2}} \right) \sigma_n(x) + \frac{2}{\sqrt{n}}, \tag{39}$$

*where $C$ is an absolute constant, and $B$ is the upper bound on the RKHS norm of $f$ given in Assumption 1.*

Lemma 5 follows from equation 7, and a probability union bound over a discretization of the domain. A detailed proof of this lemma is given at the end of this section.

Inserting the bounds on $\sigma_n(x)$ from equation 38 in Lemma 5, we have, with probability at least $1 - \delta$, uniformly in $x$,

$$|f(x) - \hat{f}_n(x)| = \mathcal{O}\left( n^{\frac{-2s+1}{2d+4s-4}} (\log(n))^{\frac{2s-1}{2d+4s-4}} (\log(\frac{n^{d-1}}{\delta}))^{\frac{1}{2}} \right), \quad \text{in the case of NT kernel,}$$

$$|f(x) - \hat{f}_n(x)| = \mathcal{O}\left( n^{\frac{-2s-1}{2d+4s}} (\log(n))^{\frac{2s+1}{2d+4s}} (\log(\frac{n^{d-1}}{\delta}))^{\frac{1}{2}} \right), \quad \text{in the case of RF kernel.}$$

That completes the proof.

*Proof of Lemma 5:*

Recall the implicit error bound given in equation 7: For a fixed $x \in \mathcal{X}$, under Assumption 1, we have, with probability at least $1 - \delta$ (Vakili et al., 2021a),

$$|f(x) - \hat{f}_n(x)| \leq \beta(\delta)\sigma_n(x), \tag{40}$$

where $\beta(\delta) = B + \frac{R}{\lambda}\sqrt{2\log(\frac{2}{\delta})}$. Lemma 5 extends this inequality to a uniform bound in $x$, using a probability union bound over a discretization of the domain.

For $f \in \mathcal{H}_k$, with $\|f\|_{\mathcal{H}_k} \leq B$, and for $n \in \mathbb{N}$, there exists a fine discretization $\mathbb{X}_n$ of $\mathcal{X}$ with size $|\mathbb{X}_n|$ such that $f(x) - f([x]_n) \leq \frac{1}{\sqrt{n}}$, where $[x]_n = \arg\min_{x' \in \mathbb{X}_n} \|x' - x\|_{l^2}$ is the closest point in $\mathbb{X}_n$ to $x$, and $|\mathbb{X}_n| \leq cB^{d-1}n^{(d-1)/2}$, where $c$ is an absolute constant independent of $n$ and $B$ (Chowdhury & Gopalan, 2017; Vakili et al., 2021a).

Under Assumption 1, it can be shown that: with probability at least $1 - \delta/2$ (Vakili et al., 2021a, Lemma 4)

$$\|\hat{f}_n\|_{\mathcal{H}_k} \leq \underbrace{B + \frac{R\sqrt{n}}{\lambda}\sqrt{2\log(\frac{4n}{\delta})}}_{U(\delta)}. \tag{41}$$

Note that $\hat{f}_n$ is a random function, where the randomness comes from the randomness in observation noise.

We thus conclude that there is a discretization $\tilde{\mathbb{X}}_n$ with size $|\tilde{\mathbb{X}}_n| \leq c(U(\delta))^{d-1}n^{(d-1)/2}$ such that $|f(x) - f([x]_n)| \leq \frac{1}{\sqrt{n}}$, and with probability at least $1 - \delta/2$,

$$|\hat{f}_n(x) - \hat{f}_n([x]_n)| \leq \frac{1}{\sqrt{n}}. \tag{42}$$

Let $\delta' = \frac{\delta}{2c(U(\delta))^{d-1}n^{(d-1)/2}}$. A probability union bound over the discretization $\tilde{\mathbb{X}}_n$ implies that: With probability at least $1 - \delta/2$, uniformly in $x \in \tilde{\mathbb{X}}_n$,

$$|f(x) - \hat{f}_n(x)| \leq \beta(\delta')\sigma_n(x). \tag{43}$$

Accounting for the discretization error from equation 42, and a probability union bound over equation 41 and equation 43, we have, with probability at least $1 - \delta$, uniformly in $x \in \mathcal{X}$,

$$
\begin{aligned}
|f(x) - \hat{f}_n(x)| &\leq |f(x) - f([x]_n)| + |f([x]_n) - \hat{f}_n([x]_n)| + |\hat{f}_n([x]_n) - \hat{f}_n(x)| \\
&\leq |f([x_n]) - \hat{f}_n([x_n])| + \frac{2}{\sqrt{n}} \\
&\leq \beta(\delta')\sigma_n(x) + \frac{2}{\sqrt{n}}.
\end{aligned}
$$

The first line holds by triangle inequality, the second line comes from the discretization error, and the third line holds by equation 43.

Inserting the value of $U(\delta) = B + \frac{R\sqrt{n}}{\lambda}\sqrt{2\log(\frac{2n}{\delta})}$ in $\delta'$, and the value of $\delta' = \frac{\delta}{2c(U(\delta))^{d-1}n^{(d-1)/2}}$ in $\beta(\cdot) = B + \frac{R}{\lambda}\sqrt{2\log(\frac{2}{\cdot})}$, we arrive at the lemma.

## I   DETAILS ON THE EXPERIMENTS

In this section, we provide further details on the experiments shown in the main text, Section 6. The code will be made available upon the acceptance of the paper.

We consider NT kernels $\kappa_{\mathrm{NT},s}(.)$, with $s = 1, 2, 3$, which correspond to wide fully connected 2 layer neural networks with activation functions $a_s(.)$. In the first step, we create a synthetic function $f$ belonging to the RKHS of a NT kernel $\kappa$. For this purpose, we randomly generate $n_0 = 100$ points on the hypersphere $\mathbb{S}^{d-1}$. Let $\hat{X}_{n_0} = [\hat{x}_i]_{i=1}^{n_0}$ denote the vector of these points. We also randomly sample $\hat{Y}_{n_0} = [\hat{y}_i]_{i=1}^{n_0}$ from a multivariate Gaussian distribution $\mathcal{N}(\mathbf{0}_{n_0}, \mathbf{K}_{n_0})$, where $[\hat{\mathbf{K}}_{n_0}]_{i,j} = \kappa(\hat{x}_i^\top \hat{x}_j)$. We define a function $g(.) = \hat{\mathbf{k}}_{n_0}^\top(.)(\hat{\mathbf{K}}_{n_0} + \delta^2 \mathbf{I}_{n_0})^{-1}\hat{Y}_{n_0}$, where $\delta^2 = 0.01$ and $[\hat{\mathbf{K}}_{n_0}(x)]_i = \kappa(x^\top \hat{x}_i)$. We then normalize $g$ with its range to obtain $f(.) = \frac{g(.)}{\max_{x \in \mathcal{X}} g(x) - \min_{x \in \mathcal{X}} g(x)}$. For a fixed $\hat{X}_{n_0}$ and $\hat{Y}_{n_0}$, $g$ is a linear combination of partial applications of the kernel. Thus $g$ is in the RKHS of $\kappa$, and its RKHS norm can be bounded as follows.

$$
\begin{aligned}
\|g\|_{\mathcal{H}_\kappa}^2 &= \left\langle \hat{\mathbf{k}}_{n_0}^\top(.)(\hat{\mathbf{K}}_{n_0} + \delta^2 \mathbf{I}_{n_0})^{-1}\hat{Y}_{n_0}, \hat{\mathbf{k}}_{n_0}^\top(.)(\hat{\mathbf{K}}_{n_0} + \delta^2 \mathbf{I}_{n_0})^{-1}\hat{Y}_{n_0} \right\rangle_{\mathcal{H}_\kappa} \\
&= \hat{Y}_{n_0}^\top(\hat{\mathbf{K}}_{n_0} + \delta^2 \mathbf{I}_{n_0})^{-1}\hat{\mathbf{K}}_{n_0}(\hat{\mathbf{K}}_{n_0} + \delta^2 \mathbf{I}_{n_0})^{-1}\hat{Y}_{n_0} \\
&= \hat{Y}_{n_0}^\top(\hat{\mathbf{K}}_{n_0} + \delta^2 \mathbf{I}_{n_0})^{-1}\hat{Y}_{n_0} - \hat{Y}_{n_0}^\top(\hat{\mathbf{K}}_{n_0} + \delta^2 \mathbf{I}_{n_0})^{-2}\hat{Y}_{n_0} \\
&\leq \hat{Y}_{n_0}^\top(\hat{\mathbf{K}}_{n_0} + \delta^2 \mathbf{I}_{n_0})^{-1}\hat{Y}_{n_0} \\
&\leq \frac{\|\hat{Y}_{n_0}\|_{l^2}^2}{\delta^2}.
\end{aligned}
$$

The second line follows from the reproducing property. We thus can see that for each fixed $\hat{X}_{n_0}$ and $\hat{Y}_{n_0}$, $g$ (and consequently $f$) belong to the RKHS of $\kappa$.

The values of $\max_{x \in \mathcal{X}} g(x)$ and $\min_{x \in \mathcal{X}} g(x)$ are numerically approximated by sampling $10,000$ points on the hypersphere, and choosing the maximum and minimum over the sample.

We then generate the training datasets $\mathcal{D}_n$ of the sizes $n = 2^i$, with $i = 1, 2, \ldots, 13$, by sampling $n$ points $X_n = [x_i]_{i=1}^n$ on the hypersphere, uniformly at random. The values $Y_n = [y_i]_{i=1}^n$ are generated according to $f$. We then train the neural network model to obtain $\hat{f}_n(.)$. The error $\max_{x \in \mathcal{X}} |f(x) - \hat{f}_n(x)|$ is then numerically approximated by sampling $10,000$ random points on the hypersphere and choosing the maximum of the sample.

We have considered 9 different cases for the pairs of the kernel and the input domain. In particular, the experiments are run for each $\kappa_{\mathrm{NT},s}$, $s = 1, 2, 3$ on all $\mathbb{S}^{d-1}$, $d = 2, 3, 4$. In addition, each one of these 9 experiments is repeated 20 times (180 experiments in total).

In Figure 2 we plot $\max_{x \in \mathcal{X}} |f(x) - \hat{f}_n(x)|$ versus $n$, averaged over 20 repetitions, for each one of the 9 experiments. Note that for $\max_{x \in \mathcal{X}} |f(x) - \hat{f}_n(x)| \sim n^\alpha$, we have $\log(\max_{x \in \mathcal{X}} |f(x) - \hat{f}_n(x)|) = \alpha \log(n) +$ constant. Thus, in our log scale plots, the slope of the line represents the exponent of the error rate. As predicted analytically, we see all the exponents are negative (the error converges to 0). In addition, the absolute value of the exponent is larger, when $s$ is larger or $d$ is smaller. The bars in the plot on the left in Figure 2, show the standard deviation of the exponents.

For training of the model, we have used *neural-tangents* library (Novak et al., 2019) that is based on *JAX* (Bradbury et al., 2018). The library is primarily suitable for $\kappa_{\mathrm{NT},1}(.)$ corresponding to the ReLU activation function. We thus made an amendment by directly feeding the expressions of the RF kernels, $\kappa_s$, $s = 2, 3$, to the *stax.Elementwise* layer provided in the library. Below we give these expressions

$$
\begin{aligned}
\kappa_2(u) &= \frac{1}{3\pi}\left[3\sin(\theta)\cos(\theta) + (\pi - \theta)(1 + 2\cos^2(\theta))\right], \\
\kappa_3(u) &= \frac{1}{15\pi}\left[15\sin(\theta) - 11\sin^3(\theta) + (\pi - \theta)(9\cos(\theta) + 6\cos^3(\theta))\right],
\end{aligned}
$$

where $\theta = \arccos(u)$.

We derived these expressions using Lemma 1 in a recursive way starting from $\kappa_1(.)$. Also, see Cho & Saul (2009), which provides a similar expression for $\kappa_2(.)$ and a general method to obtain $\kappa_s(.)$ for other values of $s$. We note that we only need to supply $\kappa_s(.)$ to the *neural-tangents* library. The NT kernel $\kappa_{\mathrm{NT},s}(.)$ will then be automatically created.

Our experiments run on a single GPU machine (*GeForce RTX 2080 Ti*) with 11 GB of VRAM memory. Each one of the 180 experiments described above takes approximately 4 minutes to run.

