# OpenReview forum: "Uniform Generalization Bounds for Overparameterized Neural Networks"
_ICLR.cc/2022/Conference — ICLR 2022 Submitted_

### Official Review · Reviewer_KpGF · 2021-10-28

**Correctness:** 4
**Technical Novelty And Significance:** 3
**Empirical Novelty And Significance:** 4
**Recommendation:** 6
**Confidence:** 3

**Main Review:**

By now, there is an established line of research that tries to understand the performance of NTK regression by understanding its associated RKHS and the decay of eigenvalues. The current papers expands on this direction in an elegant way by capturing the interplay between smoothness of activations and smoothness of learnable functions.

However, I do feel that the scope of the paper is somewhat limited. Perhaps it's not even fair to say that the paper deals with generalization bounds of overparametrized neural networks, since the main focus is on their infitely wide limit.
As the genralization bounds themselves follow from previous work, the main contribution is the equivalence between the RKHS of the Matern kernels and those of NTKs associated to the powers of ReLU. The latter activations is not very common and so the applications are restricted.

Some comments:
- The test set $\mathcal{X}$ is never fully identified. Persumably it's the sphere, but this is never mentioned explicitly in Theorems 2 and 3, which makes it hard to appreciate the result.

- The definition of the Maximal Information Gain is confusing. In Section 4 it would be a good idea to say exactly which Gaussian process is being considered and which vector Y is considered, even though their identity can be infered from the previous context. Spefically it is not clear how the parameter $\lambda$, which can be arbitrary, can be part of this definition. Perhaps it would be better to simply define it by the formula that appears at the top of page 7. This is the form which is needed at the end.

**Summary Of The Paper:**

The paper studies over-parametrized neural networks when the size of the network goes to infinity. In this regime, it is known that training the network via gradient descent type algorithms is equivalent to learning via the Neural Tangent Kernel (NTK) and the main focus is on this problem.

Previous results have shown that when one takes ReLU as the activation function, the RKHS associated with the NTK is equivalent to the RKHS of the Laplace kernel. The current paper generalizes this result to powers of ReLU and shows that as the activation function becomes smoother, so do the functions in the RKHS. Moreover, the exact decay of the eigenfunctions is also identified, in this extended case.

Using the above results the authors establish uniform bounds on the generalization error when learning a function from the appropriate RKHS with kernel regression.



**Summary Of The Review:**

The paper makes important contributions to the study of RKHS associated to the NTK. The main drawback, in my eyes, is that the scope of the paper is limited.

---

> ### Author Response · Authors · 2021-11-12
> **Response to Reviewer KpGF**
>
> We are glad that you find our contributions to the study of spectrum of neural kernels significant. We appreciate your positive feedback on the paper, and hope that you could please share it with other reviewers in the upcoming discussion. Below, we respond to your comments.
>
>
>
>
> - **The test set X**
>
> All of our results (the spectrum of neural kernels, the bounds on MIG, and the error bounds) hold on the hypersphere.
>
> Thanks for pointing out that we have missed a formal mention of this in Theorems 2 and 3. Following your suggestion, we specified the hyperspherical domain in Theorems 2 and 3, in the revised paper: "... the neural kernels with all $l\ge2$, on a hyperspherical domain $\mathcal{X}=\mathbb{S}^{d−1}$, ...".
>
>
> - **The definition of the Maximal Information Gain**
>
> It is true that formally we do not need to introduce a surrogate GP model to define the information gain. As you suggested, we can define it using the equation on top of page 7, as the log determinant of the kernel matrix (up to a scaling and adding an identity matrix). However, it is common in the literature (e.g., see, Srinivas et al., 2010; Janz et al., 2020; Vakili et al., 2021b; and references therein) to introduce a surrogate (sometimes referred to as fictitious) GP model to give some context and intuition for the information gain as the mutual information between the surrogate distributions of Y_n and F. We will be happy to further clarify this if it is not clear.
>
> We would also like to make a comment on your point on the limitation of the scope of the paper as a result of activations not being very common for $s>1$.
>
> We can mention at least two significant consequences for considering the smoother cases with $s>1$. First, we establish the equivalence between the RKHS of the neural kernels and the RKHS of the Matérn kernels with the corresponding smoothness. Matérn is perhaps the most common kernel in practice (Rasmussen and Williams, 2006; Snoek et al., 2012). Our results thus may be of general interest for kernel ridge regression and Gaussian process communities. Second, an important consequence of our results is to show that the existing regret bounds for some bandit and RL algorithms using NT kernel is non-trivial at least under some cases. In particular, as we have discussed in Section 7, typical regret bounds (see, e.g., Zhou et al., 2020; Yang et al., 2020; ZHANG et al., 2021; Gu et al., 2021) turn to sublinear only when $s>\frac{d}{2}$, which excludes the ReLU activation function.
>
> *References:*
>
> *Carl E Rasmussen and Christopher KI Williams. Gaussian Processes for Machine Learning. MIT Press, 2006.*
>
> *Jasper Snoek, Hugo Larochelle, and Ryan P Adams. Practical Bayesian optimization of machine learning algorithms. In Advances in Neural Information Processing Systems 25, 2012.*

---

### Official Review · Reviewer_kUxi · 2021-11-02

**Correctness:** 2
**Technical Novelty And Significance:** 1
**Empirical Novelty And Significance:** Not applicable
**Recommendation:** 1
**Confidence:** 5

**Main Review:**

Theorem 1 is a trivial corollary of Lemma 1 and Proposition 1. Lemma 1 is nice though it is an easy consequence of a recursive relation (4) given by Jacot et al. Proposition 1 follows directly from a nice asymptotic expression of eigenvalues of the integral operator associated with a Mercer kernel on [-1, 1] which is C^\infty on (-1, 1) but has  singularities of the same order at -1 and 1.

Theorem 2 can be easily obtained from the bound O(d_k(n) log(n)) for the maximal information gain in terms of the effective dimension d_k(n): the effective dimension d_k(n) can be easily bounded from its definition by the decay of the eigenvalues.

Theorem 3 is stated for the data tilde{D}_n = (x_i)_{i=1}^n defined in terms of the functions (sigma_i)_{i=1}^n. Since each function sigma_i is defined in terms of the original data (x_j)_{j=1}^i, the independence of the new data is not known, at least to this reviewer. Note that the function trained with the new data in Theorem 3 is different from that in (7). The proof of Theorem 3 is not trivial at all and is not included in the appendix. This reviewer believes that Theorem 3 is wrong.

In summary, Lemma 1 is a nice observation. Theorems 1 and 2 are trivial. Theorem 3 is not proved and should not be claimed as a theorem.

**Summary Of The Paper:**

The authors consider neural tangent kernels (NTKs) kappa_{NT, s}^l of the neural networks of depth l associated with ReLU a and its powers a_s with a positive integer s. Theorem 1 gives relations between the RKHSs generated by the NTKs and those generated by Matern kernels. Theorem 2 gives bounds for the maximal information gain of the NTKs. Theorem 3 provides some uniform generalization bounds for kernel ridge regression with the NTKs.

**Summary Of The Review:**

Though Lemma 1 is a nice observation, Theorems 1 and 2 are trivial. Theorem 3 is not proved and should be wrong, to this reviewer's opinion.

---

> ### Author Response · Authors · 2021-11-12
> **Response to Reviewer kUxi**
>
> We regret that you did not have as pleasant a reading as other reviewers. The criticism seems to be based on and around two points: the results are trivial, and Theorem 3 is wrong. We will address both points below.
>
> **The results are trivial and can be easily obtained:**
>
> - The statement **''Lemma 1 is nice though it is an easy consequence of a recursive relation (4) given by Jacot et al.''** is not correct. Instead, Lemma 1 is a result of two times application of Stein's lemma, a proof which is considered interesting by some other reviewers, and is not from Jacot et al., 2018. The lemma is a key step towards characterizing the spectrum of the neural (RF and NT) kernels (Proposition 1), which leads to several other substantioal results including the relation between the RKHSs of neural kernels and the RKHSs of Matérn kernels (Theorem 1), a bound on the MIG of neural kernels (Theorem 2), and a novel error bound on regression with neural kernels (Theorem 3).
>
> - We disagree with the statement that **'' the effective dimension d_k(n) can be easily bounded from its definition by the decay of the eigenvalues''**. In fact, the first bounds on the effective dimension of the most popular kernels (Matérn family, and squared exponential as a special case of Matérn family) were derived in the celebrated work of Srinivas et al., 2010 (the recipient of ICML 2020 Test of Time award). It took a decade until these bounds were improved to better bounds in Janz et al., 2020, and to optimal bounds in Vakili et al., 2021b (all these results are for the Matérn family). This shows that such results are not trivial. Although in our analysis we adopt the method of Vakili et al., 2021b, it is important to notice that the analysis of Vakili et al. 2021b does not apply to the neural kernels. The reason is that their analysis crucially relies on the assumption that the Mercer eigenfunctions are uniformly bounded. This assumption does not hold in the case of spherical harmonics. In this work, we solve this problem using Legendre *Addition* theorem, and bouding the contribution of all the spherical harmonics of a certain degree to the kernel by Legendre polynomials. Please see the details in Section 4.2 and Appendix G.
>
> - Our bounds on MIG (and consequently on the effective dimension) may be extremely important for researchers across several communities. There is a growing literature (e.g., Zhou et al., 2020; Yang et al., 2020; ZHANG et al., 2021; Gu et al., 2021) leveraging neural kernels to provide analytical bounds for RL and bandit problems which use certain neural network modelling. We believe that our bounds on MIG are crucial for this line of research. For example, see row 6 in Table 1 of Yang et al., 2020, where the regret bound is given in terms of the effective dimension ($d_{eff}$, in their notation), without characterizing it. Their result thus seems incomplete. Our bounds on MIG, and consequently on the effective dimension ($\tilde{d}_k(n)$ in our notation) can be easily inserted in their results to obtain a complete version of the regret bound explicit in $n$ ($T$, in their notation). This also shows that the results are not trivial. In addition, it shows that the results are useful for other researchers.
>
> **The proof of Theorem 3 is not trivial at all and is not included in the appendix. This reviewer believes that Theorem 3 is wrong.**
>
> As stated in the paper, the proof follows the same steps as in the proof of Theorem 3 of Vakili et al. (2021a). Their theorem holds for general kernels, provided the bound on MIG. We have adopted their theorem for the special case of neural kernels, inserting our novel bounds on MIG of the neural kernels. Following your comment, for completeness, we added the proof (please see Appendix H in the revised version). Please let us know if something is unclear, and we will be happy to discuss.
>
> *Additional reference:*
>
> *Janz, David, David Burt, and Javier González. "Bandit optimisation of functions in the Matérn kernel RKHS." International Conference on Artificial Intelligence and Statistics. PMLR, 2020.*

---

### Official Review · Reviewer_Enyd · 2021-11-02

**Correctness:** 4
**Technical Novelty And Significance:** 3
**Empirical Novelty And Significance:** 2
**Recommendation:** 6
**Confidence:** 2

**Main Review:**

The paper is well-written and the results make non-trivial extensions over the state-of-the-art. That being said, it is worth pointing out that the work is mostly of purely mathematical interest (which is fine): the case of ReLU activation functions was already treated in earlier work. Nevertheless, the extra difficulty involved in the case here is what motivates the use of Stein's Lemma, which makes the paper (moderately) interesting from the technical point of view. The exposition of the concepts is very good in general, though some references and some (minor) aspects appear to be missing.  Overall, I am happy to recommend acceptance.

Below are some more specific comments.


// =======after rebuttal======
After reading the other reviews and thee rebuttal, I would incline to slightly lower my score to (6).

I notice that reviewer gYma and I both complained about the general context of "uniform generalization bounds". Contrary to reviewer gYma, I don't think the restrictions imposed make the paper unworthy of publication in itself. However, I believe the authors should try harder to explain the enormous difference between the setting they study and classic "generalization bounds". The rebuttal is slightly dismissive in that respect.

More importantly, the issue pointed out by reviewer gYma about Theorem 3 is quite serious. The proof the authors added is not that short, which indicates the paper was somewhat rushed in terms of making the deadline.

I still stand with the authors in terms of originality, but with low confidence.

In conclusion, this remains a valuable paper from the purely mathematical point of view (with limited practical applicability, though that is not of great importance), but some parts are a bit rushed and the placement of the paper within the broader literature and other pedagogical aspects are somewhat neglected. I think it is still above the borderline but not by as far as I originally thought.


/
======================
General comments/questions about the content
======================

1. On page 6, when it is claimed that the relevant kernels are equivalent to a Matérn family of kernels, it would be nice to point the reader to the definition of the "equivalence" between kernels (which is at the beginning of the supplementary), since otherwise it sounds a bit like the two learning problems are completely equivalent (which is not the case).


2. The part of the related works which deals with "Classical approaches" (in the introduction and in the rather bizarre Section C) is not very complete. It is really not fair for the present paper to present itself as one of the only ones that exploited the NTK regime for generalization purposes! At the absolute least it is imperative to cite [4]. In a more general discussion of pre-NTK work it would be natural to cite [3] as well, though it is less required,

3. On a related note, it should be made much clearer in the "generalization bounds" part of the paper that the rate is worse than existing rates because of the lack of an i.i.d. assumption: a rate of $1/\sqrt{n}$ is pretty much known under such assumptions.

4. In page 15, I really appreciate the fact that the exact result from [1] was written again here. To make things even better, it would be nice to explain that one can construct a rotationally invariant kernel $k$ from $\kappa$ via $k(x,y)=\kappa(y^\top x)$, rather than simply saying "let $\kappa:[-1,1]\rightarrow \mathbb{R}$ be a rotationally invariant kernel. I know these concepts are explained in the main paper but it doesn't hurt to help the reader a bit.


5. The RL connection is slightly oversold.


6. It is rather strange that you are not citing any paper when using Stein's lemma. There is a whole branch of literature that uses Stein's lemma in many different areas of statistics and probability. Relevant references are [5,6,7] to name but a few.






/
===================
(Very) Minor Comments/typos (non exhaustive)
===================



1.  Around the middle of page 2: "That we show to imply..." =====> "Then we show this implies..."

2. Bottom of page 4: "...share the same eigenvalue $\tilde{\lambda}$, is $N$..." ====> "...share the same eigenvalue $\tilde{\lambda}$, i.e. $N$..."

3. Middle of page 5: "They derived those expansions for the case of ReLU activarion function." ===> "They derived those expansions for the case of the ReLU activarion function."

4. Beginning of Section 3.2 on page 6: "Our bounds...shows" ====> "Our bounds...show"


5. Bottom of page 6: it would be nice to say (at least once) that "GP" stands for "Gaussian Process".

6. Beginning of section 5.2 on page 8: "provide explicit bounds on error" ====>"provide explicit bounds on the error"

7. Discussion (around the middle) on page 9: "...the corresponding eigenvalues. our..." ====>  "...the corresponding eigenvalues. Our..."

8. Beginning of Section B in the supplementary (page 13): "we overview the Mercer's Theorem" ====>"we give an overview of Mercer's Theorem"

9. Below the main equations in Section D (proof of Lemma 1): "applying the Stein's Lemma"===> "applying Stein's Lemma"


























/
==================
References
==================

[1] Alberto Bietti and Francis Bach. Deep equals shallow for ReLU networks in kernel regimes. ICLR 2021.

[2] Amnon Geifman, Abhay Yadav, Yoni Kasten, Meirav Galun, David Jacobs, and Basri Ronen. On the similarity between the Laplace and neural tangent kernels. NeurIPS 2020.

[3] Peter L. Bartlett, Dylan J. Foster, Matus J. Telgarsky. Spectrally-normalized margin bounds for neural networks. NeurIPS 2017.

[4] Sanjeev Arora, Simon S. Du, Wei Hu, Zhiyuan Li, Ruosong Wang.  Fine-Grained Analysis of Optimization and Generalization for Overparameterized Two-Layer Neural Networks. ICML 2019.

[5] Charles Stein. Approximate computation of expectations. Institute of Mathematical Statistics Lecture Notes, Monograph Series, 7. 1986.

[6] Louis H.Y. Chen, Larry Goldstein, Qi-Man Shao. Normal Approximation by Stein’s Method. Probability and Its Applications. 2011.

[7] Ivan Nourdin & Giovanni Peccati. Stein’s method on Wiener chaos. Probability Theory and Related Fields. 2009.






**Summary Of The Paper:**

The main contribution of this paper is to characterise the eigenvalue decay of the Neural Tangent Kernels associated with fully-connected neural networks evaluated on the hypersphere when the activation functions of the form $Relu^{s}$ where $s$ is a parameter that regulates the smoothness of the activation functions. The results generalise existing results for the particular case $s=1$, i.e. the case of ReLU activations [1,2]. The idea of the proof is to rely on a recent result from [1] which bounds the eigenvalye decay of an arbitrary rotationally invariant kernel based on smoothness and end-point asymptotic properties of the kernel written as a function from $[-1,1]$  to $\mathbb{R}$ where $[-1,1]$ corresponds to the inner product between the two arguments of the kernel function. The computation of the  asymptotics relies on an elegant use of Stein's Lemma as well as the recurrent relation between the NTK and RFK (random feature kernel) of various layers. Results for the eigenvalue decay of the RFK are also provided. Similarly to other works, once eigenvalue decay is obtained, various results can be obtained which further characterise the kernel and the learning problem: (1) it is shown that the kernel is equivalent to a Matérn family of kernels (which itself is equivalent to a sobolev space approach), (2) data-dependent generalisation bounds are shown where the distance between the empirical estimate and the groudn truth is bounded by a term that depends on the relationship between the test point and the training points (3) generalization bounds in $L^\infty$ norm with some strong assumption on the training points (they essentially have to be chosen in a "perfect" way). Applications to reinforcement learning are hinted at in a discussion at the end.




/
==================
References
==================

[1] Alberto Bietti and Francis Bach. Deep equals shallow for ReLU networks in kernel regimes. ICLR 2021.

[2] Amnon Geifman, Abhay Yadav, Yoni Kasten, Meirav Galun, David Jacobs, and Basri Ronen. On the similarity between the Laplace and neural tangent kernels. NeurIPS 2020.




**Summary Of The Review:**

This is a well-written paper with a non trivial advance over the state of the art and a reasonably large number of computations, with one semi-novel technique involved.

---

> ### Author Response · Authors · 2021-11-12
> **Response to Reviewer Enyd**
>
> Thanks for your thorough review of our paper and detailed comments. We are very happy to see your positive view on the paper, and hope that you could please share it with other reviewers in the upcoming discussion. Below we will respond to your comments in detail.
>
> **General comments/questions about the content**
>
>
> 1. **"equivalence" between RKHSs.**
>
> We have given the mathematical definition of the equivalence between two spaces (here the RKHSs) in Appendix A. Following your suggestion, we commented on that right after Theorem 1.
>
> 2. **References**
>
> Following your suggestion, we added a reference to [3].
>
> 3. **On a related note, it should be made much clearer in the "generalization bounds" part of the paper that the rate is worse than existing rates because of the lack of an i.i.d. assumption: a rate of 1/n is pretty much known under such assumptions.**
>
> We would like to emphasize that in contrast to *average* or $L^2$ norm of the error, we provide error bounds under a much stronger criteria, that is the absolute ($L^\infty$) error. Please see the second paragraph in the Discussion section.
>
> We are not aware of any $L^\infty$ result under i.i.d. case comparable to ours. We appreciate if you could please let us know if we are missing something here.
>
> 4. **In page 15, I really appreciate the fact that the exact result from [1] was written again here. To make things even better, it would be nice to explain that one can construct a rotationally invariant kernel k from κ via k(x,y)=κ(y⊤x), rather than simply saying "let κ:[−1,1]→R be a rotationally invariant kernel. I know these concepts are explained in the main paper but it doesn't hurt to help the reader a bit.**
>
> Following your suggestion, we added a comment right before Lemma 3 on page 15.
>
> 5. **The RL connection is slightly oversold**.
>
> There is a growing literature (e.g., Zhou et al., 2020; Yang et al., 2020; ZHANG et al., 2021; Gu et al., 2021) leveraging neural kernels to provide analytical bounds for RL and bandit problems which use certain neural network modelling. We believe that our bounds on MIG are crucial for this line of research. For example, see row 6 in Table 1 of Yang et al., 2020, where the regret bound is given in terms of the effective dimension ($d_{eff}$, in their notation), without characterizing it. Their result thus seems incomplete. Our bounds on MIG, and consequently on the effective dimension ($\tilde{d}_k(n)$ in our notation) can be easily inserted in their results to obtain a complete version of the regret bound explicit in $n$ ($T$, in their notation).
>
> 6. **It is rather strange that you are not citing any paper when using Stein's lemma. There is a whole branch of literature that uses Stein's lemma in many different areas of statistics and probability. Relevant references are [5,6,7] to name but a few.**
>
> Thanks, following your suggestion, we added references for Stein's lemma.
>
> **Minor Comments/typos**
>
> The abbreviation GP is introduced in Section 2.1, where the term Gaussian process first appears.
>
> We fixed all other typos.
>
> Thank you for your careful reading of the paper and detailed comments. We apprecite the amount of time that is gone into this review.

---

### Official Review · Reviewer_gYma · 2021-11-02

**Correctness:** 2
**Technical Novelty And Significance:** 1
**Empirical Novelty And Significance:** 1
**Recommendation:** 3
**Confidence:** 4

**Main Review:**

## Strength:

This is a very well-written and coherent paper.

## Weaknesses:

**Theorem 3**: The authors assume $D_n = \set{x_i,y_i}, i < n$   is constructed using a special non-i.i.d sampling procedure often referred to as Uncertainty Sampling [1] within the field of Active Learning.  Adopting the Bayesian prior assumption  $f \sim GP(0,NTK) , x_i = \underset{{x \in X}}{argmax} \\  \sigma_{i-1}(x) $ are picked by maximizing the posterior variance of $f$.

A number of comments regarding the relevance and novelty of Theorem 3 follows.

1. A proof of this theorem is not given in the main text or the appendix, hence I am not able to verify the correctness of the statement.

2. The assumption on the dataset does not apply to any supervised learning task with i.i.d data.

3. Without any assumptions on the dataset, Theorem 3.2 of [2] proves a similar statement. They show that for a fixed x, $| f_{ntk}(x)- \hat f(x) | < Poly(n, log(1/\delta))$ with probability greater than $1- \delta$. Here, $f_{ntk}$ is the kernel regression predictor, using the NTK, on the same dataset, and $\hat f(x)$ is an MLP trained using gradient descent on dataset $D_n$. Using a basic least squares error bound for kernel regression, e.g. Theorem 13.5 [3], and the triangle inequality, one can produce a bound similar to Theorem 3 of this paper, which holds for any fixed dataset, without further restrictive assumptions on how the data needs to be acquired. My point here is that, to me this bound deems incremental taking into account the restrictive problem setting, and considering the existence of powerful results such as Theorem 3.2 [2], or Section 5 of [4].

4. This theorem may be useful as a lemma within the literature of Bandit Optimization and Active Learning, however Lemma 4.2 [6] and Lemma 5.2 [5] establish very similar results under less restrictive assumptions on the dataset. These papers have no assumption over how $x_i$ are collected and only assume that $y_i$ are independent, conditioned on $x_i$.

5. Lastly, I find the title uniform generalization bound to be somewhat deceiving, as Theorem 3 holds with high probability for a fixed $f \in \mathcal{H}_k$, and not uniformly for all members of this RKHS. A uniform bound (e.g. Glivenko Cantelli-type theorems) is often referred to as a bound that holds simultaneously for all functions in the hypothesis class. I can not see how this bound can be extended to hold uniformly over $\mathcal{H}_k$.


**Experiments**: The code for the experiments was not given by the authors, so I can't verify the correctness of the experiments. Also, theorem 3 assumes the dataset is collected using a non-i.i.d sampling procedure often referred to as uncertainty sampling. However, in the experiments the authors collect the data randomly which breaks their assumption. This inconsistency is not explained in the paper.

**Other contributions**: As stated by the paper, the presented results for Theorem 1 already exist for $s = 1$ in [7], and this paper contribution is to generalize it for $s > 1$. The techniques used for generalizing to $s > 1$ are not novel, and it is the same used as in [7] and [8]. Although these results could have an interesting theoretical value, it is not practical since DNNs do not use activation function  for $s > 1$  as correctly stated by the authors in the Remark 2.

**Literature review**: Theorem 2 of this paper is an extension of Theorem 3.1 of [6] which bounds the MIG for $s=1$ to the $s >1$ case, however there is no reference to [6]. Moreover, Lemma 4.2 of [6] presents a similar generalization bound for MLPs for a general i.i.d dataset in the case of $s=1$, yet despite the similarity it is not mentioned.





**Summary Of The Paper:**

In this paper, the authors provide a more generalized equivalence of the NT and NNGP (referred as RF in this paper) kernels of MLPs and Matern kernels for activation functions of form $max(0,x)^s$ by analysing the eigen decay of the kernels corresponding Gram matrices. Later they use this eigen decay to bound the maximal information gain (MIG) of the kernels and finally present a generalization bound with respect to the number of the samples.

**Summary Of The Review:**

I think the contributions of this paper are incremental, and is limited to using already known techniques to generalize results in [7] and [6] for $s > 1$. Another contribution is to derive the bounds in Theorem 1, 2 and 3 for RF kernels in addition to NT kernels. However the results for RF kernels are implicit in the other works.

## References

[1] David D Lewis and Jason Catlett.  Heterogeneous uncertainty sampling for supervised learning.  In Machine learning proceedings 1994, pp. 148–156. Elsevier, 1994.

[2] Sanjeev Arora, Simon S Du, Wei Hu, Zhiyuan Li, Ruslan Salakhutdinov, and Ruosong Wang.  On exact computation with an infinitely wide neural net. arXiv preprint arXiv:1904.11955, 2019.

[3] Martin J Wainwright. High-dimensional statistics: A non-asymptotic viewpoint , volume 48.  Cambridge University Press, 2019.

[4] Arthur Jacot, Franck Gabriel, and Clement Hongler.  Neural tangent kernel: Convergence and generalization in neural networks. arXiv preprint arXiv:1806.07572, 2018.

[5] Dongruo  Zhou,  Lihong  Li,  and  Quanquan  Gu.   Neural  contextual  bandits  with  UCB-based  exploration.    In  Hal  Daume  III  and  Aarti  Singh  (eds.),Proceedings of the 37th International Conference on Machine Learning, volume 119 of Proceedings of Machine Learning Research,pp. 11492–11502. PMLR, 13–18 Jul 2020.

[6] Kassraie, Parnian, and Andreas Krause. "Neural Contextual Bandits without Regret." arXiv preprint arXiv:2107.03144 (2021).

[7] Geifman, Amnon, et al. "On the similarity between the laplace and neural tangent kernels." arXiv preprint arXiv:2007.01580 (2020).

[8] Alberto Bietti and Francis Bach. Deep equals shallow for ReLU networks in kernel regimes. arXiv preprint arXiv:2009.14397, 2020.

---

> ### Author Response · Authors · 2021-11-12
> **Response to Reviewer gYma (Part 1)**
>
> We are glad that you find the paper very well-written and coherent, and we regret that some minor issues and misconceptions have negatively affected your view on the paper. Also, we appreciate the amount of time that is gone into this review, and hope to address all of your comments.
>
> To address your comments in detail, we have separated the answers into two parts: answers to points 1, 2 and 3 on Theorem 3 in this part, and answers to the rest of your comments in the next part.
>
>
> **Comments regarding the relevance and novelty of Theorem 3:**
>
> 1. **Proof of Theorem 3**
>
> As stated in the paper, the proof follows the same steps as in the proof of Theorem 3 of Vakili et al. (2021a). Their theorem holds for general kernels, provided the bound on MIG. We have adopted their theorem for the special case of neural kernels, inserting our novel bounds on MIG of the neural kernels. Following your comment, for completeness, we added the proof (please see Appendix H of the revised paper). Please let us know if something is unclear, and we will be happy to discuss.
>
> 2. **i.i.d. data**
>
> We agree with the reviewer that our assumption (of *uncertainty sampling*) on dataset does not apply to i.i.d. data. We would however like to argue that this assumption is not necessarily more limiting than the i.i.d. data assumption (sampling from a prior distribution), that is often employed in similar results on error bounds.
>
> Assuming that the data is sampled from a prior distribution is an analytical tool (similar to uncertainty sampling) which is sometimes employed for convenience or feasibility of analysis. Such assumption also sometimes does not have a basis in reality. In fact, assuming a prior suitable for the analysis is also often criticized on more philosophical grounds which are rooted in the classic divide between Bayesian and frequentist statistics.
>
> When sampling from a prior distribution on the hypersphere, often a uniform distribution is considered (e.g., Wang et al. (2020), Bordelon et al. (2020) ) to ensure a quasi-uniform dataset, evenly distributed across the domain. Interestingly, uncertainty sampling also results in a quasi-uniformly distributed dataset under mild assumptions on the kernel (Wenzel et al. 2021). Thus, overall the two approaches do not seem inherently different. They are just suitable for different analytical techniques.
>
> One possible direction to follow here is to extend the scope of our Theorem 3 to all quasi-uniformly distributed datasets (where the max distance between the input points is bounded by $O(n^{-\frac{1}{d-1}})$ on the sphere). That however seems to need different analytical tools. We acknowledge that (only) our Theorem 3 crucially relies on uncertainty sampling. The direction mentioned here may be interesting for future investigations.
>
> 3. **Theorem 3.2 of [2], Theorem 13.5 [3]**
>
> - **"Without any assumptions on the dataset, Theorem 3.2 of [2] proves a similar statement"**: Theorem 3.2 of [2] is in nature different from our result. That theorem is concerned with bounding the error between the NT kernel model and finite width networks (with sufficiently large width). That is complementary to our result.
>
> - The statement that **"Using a basic least squares error ... how the data needs to be acquired"** is not correct. Theorem 13.5 of [3] is on a general case of nonparametric regression bounds based on Gaussian complexity. The same book gives the specification of their Theorem 13.5 for RKHSs, in their Theorem 13.17, which facilitates a clear comparison to our results. (Please let us know if we should not compare with 13.17 instead of 13.5). In both of those theorems, the *empirical $l^2$ norm* of the error is considered, which is a much weaker criteria than our uniform ($L^\infty$) error bound. Although the bounds on the *empirical $l^2$ norm* of the error are fundamental results and of independnet interest, they do not imply our resutls. Please let us know if we are missing something, and we will be happy to discuss.
>
> - On the previous item, we also note that, with no assumption on the dataset (e.g., uncertainty sampling or i.i.d.), the error cannot converge to zero uniformly (in x). As we have mentioned in Section 5.2: "It is clear that with no assumption on the distribution of the data, generalization error cannot be nontrivially bounded.  For example, if all the data points are collected from a small region of the input domain, it is not expected for the error to be small far from this small region".

---

> ### Author Response · Authors · 2021-11-12
> **Response to Reviewer gYma (Part 2)**
>
> **Comments regarding the relevance and novelty of Theorem 3:**
>
>
> 4. **Lemma 4.2 [6] and Lemma 5.2 [5] establish very similar results**
>
> We disagree with this assessment. Our Theorem 3 is significantly different from Lemma 4.2 of [6] and Lemma 5.2 of [5], which are **implicit** error bounds. Please notice that Lemma 4.2 of [6] and Lemma 5.2 of [5] are similar to our equation (7) that is from (Vakili et al., 2021a). While, our Theorem 3 combines such implicit error bounds with our novel bounds on MIG of the neural kernels to prove explicit in $n$ error bounds. Please see more details below.
>
> - Lemma 4.2 of [6] and Lemma 5.2 of [5] are extensions of self-normalized martingale inequalities, first introduced in the context of linear bandits (Abbasi-Yadkori et al. 2011), to linear regression in the feature space of neural networks. They also use Theorem 3.2 of [2], which you mentioned above, to account for the error due to width.
>
> - In contrast, equation (7) that is from (Vakili et al., 2021a) is specific to the kernel regime. That however is stronger than the results in Lemma 4.2 of [6] and Lemma 5.2 of [5], in removing an $O(\sqrt{\gamma_k(n)})$ from the confidence interval width. For a detailed discussion on this improvement, please see Section 3 of (Vakili et al., 2021a). This $O(\sqrt{\gamma_k(n)})$ difference is in particular the reason behind the fact that the regret bounds in bandit and RL problems are not necessarily sublinear. See Section 4 of (Vakili et al., 2021a) for the details.
>
> - In the paper, we have referred to equation (7) that is from (Vakili et al., 2021a) as an **implicit** error bound. It is implicit in the sense that it gives the error in terms of an uncertainty estimate depending on data. That is the same for Lemma 4.2 of [6] and Lemma 5.2 of [5]. Indeed, this kind of implicit error bounds apply to all data sets.
>
> - Our Theorem 3 uses implicit error bounds only as one step. Two important components which allow our explicit error bounds which does not exist in the previous work and go beyond the implicit error bounds are  i) our novel bound on the MIG of neural kernels in Theorem 2 (which is based on our characterization of the spectrum of neural kernels in Proposition 1). ii) The choice of our dataset based on an *uncertainty reduction* design. Theorem 3 is a result of these components which absolutely do not exist in Lemma 4.2 of [6] and Lemma 5.2 of [5].
>
> 5. **Uniform generalization bound**
>
> We use *uniform* to refer to uniform, in $x$, error bounds. Meaning, our bounds hold uniformly across the domain (the hypersphere here). This is important because most exiting results are on average or $L^2$ bounds on the error. We believe this is a standard terminology for convergence of two functions to each other (here $f$ and $\hat{f}_n$), uniformly over the domain. Please let us know if this is not clear, and we will be happy to add a comment in the paper.
>
>
> **Experiments**
>
> The code is now available: https://github.com/oppiropaper/submission/tree/main
>
> As discussed above, both approaches result in quasi-uniform datasets and show the same performance when $n$ is relatively large, as confirmed with our experiments. For large scale experiments we have collected the data randomly since it is much faster.
>
> **Literature Review**
>
> For Lemma 4.2 of [6], please see the response above. Thanks for pointing out [6], which is recently posted to *arXiv*. We added a reference to it, and made it clear that our Theorem 2 is a generalization of Theorem 3.1 of [6].
>
> Thanks again for your detailed review. Looking forward to your feedback on our response.
>
> *References:*
>
> *Wenzel, Tizian, Gabriele Santin, and Bernard Haasdonk. "A novel class of stabilized greedy kernel approximation algorithms: Convergence, stability and uniform point distribution." *Journal of Approximation Theory* 262 (2021): 105508.*
>
> *Abbasi-Yadkori, Yasin, Dávid Pál, and Csaba Szepesvári. "Improved algorithms for linear stochastic bandits." *Advances in neural information processing systems* 24 (2011): 2312-2320.*

---

### Official Review · Reviewer_V82Y · 2021-11-05

**Correctness:** 4
**Technical Novelty And Significance:** 4
**Empirical Novelty And Significance:** Not applicable
**Recommendation:** 6
**Confidence:** 2

**Main Review:**

The presented uniform generalization bound for kernel ridge regression in the reproducing kernel Hilbert spaces (RKHSs) generated by neural tangent kernels and the maximum information gain (MIG) approach are of interest. The first theorem is about the equivalence of the RKHSs of the neural tangent and Matern kernels over the hyperspherical domain. To bound the generalization error, the authors provide a bound on the MIG of neural tangent kernels and random feature kernels in Theorem 2. Then convergence rates for the generalization error are presented by considering a special data collection module. Overall, the paper is well-written and technically sound. I vote for acceptance. I have some concerns as follows:

1. The assumption that the approximation error is zero is kind of strong.

2. The benefit of using ReLU with $s>1$ is not so clear to me. Is it possible to consider other more general activation functions and derive similar MIG bounds?

3. In Section 6, why choose $\hat{Y}\subset\mathbb{R}^d$? Is $d$ the dimension of the input space? It is also a little bit strange that you let $f$ depend on $\hat{k}_{n_0}$, i.e., $s$, when you are trying to compare the error exponent over $s$.

4. Are the rates derived in Theorem 3 minimax optimal?

**Summary Of The Paper:**

The paper provides a uniform generalization bound for overparameterized neural networks by assuming that the target function resides in the reproducing kernel Hilbert spaces generated by Neural Tangent (NT) kernels associated with general ReLU activation functions.  The approach is based on so-called maximum information gain, which is neat. The equivalence of the RKHSs generated by NT kernels and Matern kernels are discussed.

**Summary Of The Review:**

The approach based on maximum information gain is very interesting. The uniform generalization bound for overparameterized neural networks given in this paper is neat.

---

> ### Author Response · Authors · 2021-11-12
> **Response to Reviewer V82Y**
>
> Thank you for your positive feedback on our work. We are glad that you find the paper well written, and its contributions interesting. We also believe that the results may be of interest for researchers across several communities. We hope that you could please share your interest in the paper with other reviewers in the upcoming discussion. Below we will respond to your itemized comments.
>
>
> 1. **The assumption that the approximation error is zero is kind of strong.**
>
> Our work falls within the neural (NT and RF) kernels framework. As mentioned by Reviewer KpGF, "by now, there is an established line of research that tries to understand the performance of NTK regression" using the spectrum of the kernel. We believe, our work makes significant contributions to the literature by i) characterizing the spectrum of the neural kernels under a general case, ii) establishing bounds on the MIG of the neural kernels, iii) and providing novel explicit bounds on a regression problem. We however agree with the reviewer that the approximation error of the NT kernel framework is an important problem. Analysis of finite width neural networks is beyond the scope of this paper, and is complementary to our results.
>
> 2. **The benefit of using ReLU with $s>1$ is not so clear to me. Is it possible to consider other more general activation functions and derive similar MIG bounds?**
>
> We can mention at least two significant consequences for considering the smoother cases with $s>1$. First, we establish the equivalence between the RKHS of the neural kernels and the RKHS of the Matérn kernels with the corresponding smoothness. Matérn is perhaps the most common kernel in practice (Rasmussen and Williams, 2006; Snoek et al., 2012). Our results thus may be of general interest for kernel ridge regression and Gaussian process communities. Second, an important consequence of our results is to show that the existing regret bounds for some bandit and RL algorithms using NT kernel is non-trivial at least under some cases. In particular, as we have discussed in Section 7, typical regret bounds (see, e.g., Zhou et al., 2020; Yang et al., 2020; ZHANG et al., 2021; Gu et al., 2021) turn to sublinear only when $s>\frac{d}{2}$, which excludes the ReLU activation function. In response to your question, our methodology can be extended to other activation functions, when the spectrum of the neural kernels can be characterized through the endpoint expansions of the kernels (as in our Proposition 1).
>
> 3. **In Section 6, why choose $\hat{Y}\subset R^d$? Is $d$ the dimension of the input space? It is also a little bit strange that you let $f$ depend on $\hat{k}_{n_0}$, i.e., $s$, when you are trying to compare the error exponent over $s$.**
>
> There is a typo: we meant $\hat{Y}_{n_0}\in \mathbb{R}^{n_0}$, that is an arbitrary vector whose elements are in $\mathbb{R}$. The typo is now fixed. Thanks for pointing this out.
>
> Our choice of $f$ ensures that it belongs to the RKHS of the corresponding kernel. We prove this claim by providing a bound on the RKHS norm of $f$ in the Appendix. On choosing $f$ based on $s$, please notice that $s$ determines the kernel, thus the RKHS. And, we want $f$ to belong to the RKHS. This is a common way of creating functions in an RKHS (e.g., see the experiments section of Chowdhury and Gopalan, 2017). We reiterate that our error bounds hold under the assumption that $f$ belongs to the corresponding RKHS.
>
> 4. **Are the rates derived in Theorem 3 minimax optimal?**
>
> We are not sure. Proving a lower bound may be a more challenging problem.
>
> *References:*
>
> *Carl E Rasmussen and Christopher KI Williams. Gaussian Processes for Machine Learning. MIT Press, 2006.*
>
> *Jasper Snoek, Hugo Larochelle, and Ryan P Adams. Practical Bayesian optimization of machine learning algorithms. In Advances in Neural Information Processing Systems 25, 2012.*
>
> *Sayak Ray Chowdhury and Aditya Gopalan. On kernelized multi-armed bandits. In International Conference on Machine Learning, 2017.*

---

### Decision · Program_Chairs · 2022-01-20

**Decision:**

Reject

**Comment:**

The paper provides a uniform generalization bound for overparameterized neural networks using the notion of maximal information gain. The analysis relies on the eigendecay of the eigenvalues of the NTK, which has recently been the object of a lot of work in the literature, including the work of Bietti and Bach (the proof actually uses one of their key lemma).

The paper originally received a set of reviews with a large disagreement between the reviewers (including two reviewers with a negative opinion and three reviewers being more positive). After the discussion period, two reviewers kept a very negative opinion, while other reviewers slightly lowered their score. Some of the problems raised by the reviewers include the restrictions imposed on the data, a missing proof (which was eventually added by the authors), the discussion of prior work being inadequate (including for instance the differences with more classical generalization bounds), and the novelty of the analysis.

Overall, the paper clearly has some merits but some of the concerns above are too important at this stage to accept the paper. I recommend the authors address the concerns mentioned in the reviews before re-submission.